# Quantification and Control of LSTM Resilience Based on Stability Theory

## Abstract

This paper proposes a novel theoretical framework for guaranteeing and evaluating the resilience of long short-term memory (LSTM) networks in control systems. We introduce *recovery time* as a new metric of resilience in order to quantify the time required for an LSTM to return to its normal state after anomalous inputs. By mathematically refining incremental input-to-state stability ($\delta$ISS) theory for LSTM, we derive a practical data-independent upper bound on recovery time. This upper bound gives us resilience-aware training. Experimental validation on simple models demonstrates the effectiveness of our resilience estimation and control methods, enhancing a foundation for rigorous quality assurance in safety-critical AI applications.

## 1 Introduction

We introduce the background, objectives, and contributions of this study.

### 1.1 Background

In recent years, artificial intelligence (AI) has become increasingly integrated into various aspects of our daily lives, from healthcare to autonomous vehicles. This widespread adoption has raised concerns regarding AI system reliability and highlighted the need for robust quality assurance measures. In response to these trends, multiple organizations and regulatory bodies have published guidelines for ensuring AI safety and reliability, such as the European Union's AI Act (European Parliament and Council, 2024) and AI HLEG guidelines (High-Level Expert Group on Artificial Intelligence, 2019). Although these guidelines have contributed to the gradual establishment of AI quality assurance processes, the development of specific evaluation metrics and implementation methodologies remains in its nascent stages. These are critical issues, especially for control systems incorporating AI, because such systems interact directly with the physical world, potentially impacting human lives in extreme scenarios.

In this paper, we focus on the control of time-series models, particularly long short-term memory (LSTM) networks, having real-time capability, which is an important property in control systems. While typical time-series models can extract nonlinear features from time-series data, store them as internal states, and utilize them to make inferences at each time step, LSTM surpasses other models in the accuracy of prediction (see, e.g., Gers et al., 2002; Ma et al., 2015; Zhao et al., 2017). This ability is well-suited for predicting future outputs of nonlinear systems in model predictive control (MPC), and therefore LSTM's capacity to track target values is better than that of traditional control methods (see, e.g., Wu et al., 2019; Chen et al., 2020; Kang et al., 2023). Moreover, LSTM can be implemented on Field Programmable Gate Array (FPGA), which is suitable for machine embedding (Guan et al., 2017). However, the state-space nature of time-series models also presents significant risks: If a model receives temporarily anomalous input values, it may retain inaccurate state memories, potentially leading to persistent errors in prediction that could significantly compromise the system's safety and reliability.

The concerns arising from the use of time-series models for control systems can be mitigated by ensuring stability, robustness, and resilience in these models (see, e.g., Dawson et al., 2022). Stability refers to

consistent performance across similar inputs, which forms the foundation for robustness and resilience. Robustness refers to tolerance against perturbations, while resilience refers to the ability to return to a normal state after transitioning to an abnormal mode due to anomalous inputs. See also Section 2.1 for related work on these concepts. Time-series models with these properties should exhibit consistent behavior under normal conditions. However, these properties cannot be adequately evaluated using conventional metrics focused on inference accuracy, such as precision, recall, and F1-score (Sengupta & Friston, 2018). Therefore, it is necessary to formulate new evaluation metrics and develop techniques to achieve the required metrics. This issue might be addressed by comprehensively collecting and learning from data across various scenarios for evaluation, but there are difficulties in defining comprehensiveness and collecting data at scale. Given this context, we mathematically formulate the resilience of LSTM, which is different from data-driven approaches. For theoretical and experimental approaches in assurance of LSTM stability and quality, see also Section 2.2 and Section 2.3.

### 1.2 Objective

Our research focuses on formulating evaluation metrics and implementation methodologies to assess and ensure the resilience of LSTMs. Specifically, our goals are as follows:

- To introduce a novel mathematical formulation of the resilience, *recovery time*, defined as the duration required for a system to return to its normal state after transitioning to an abnormal mode due to anomalous inputs (see Figure 1).

- To theoretically evaluate the upper bound of recovery time and to develop a practical data-independent method for estimating it.

- To present parameter adjustment techniques for controlling the recovery time of the model during the training process.

The key tool to achieve our goals is the incremental input-to-state stability ($\delta$ISS) (Angeli, 2002; Bayer

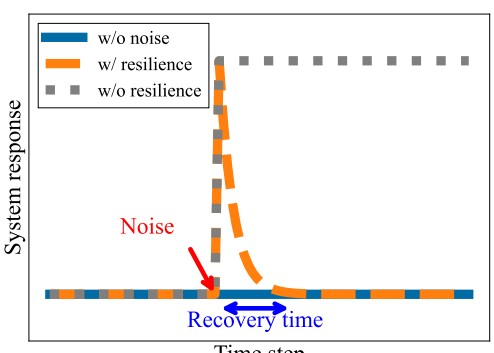

Figure 1: Schematic diagram of recovery time. The blue line depicts the system's response to normal input without perturbations, while the orange and gray lines illustrate responses of systems with and without resilience to anomalous inputs subjected to instantaneous noise, respectively. We define the recovery time as the interval required for a system's response to revert to its nominal behavior following a perturbation induced by noise. A resilient system is a system that has a finite recovery time.

et al., 2013), because $\delta$ISS guarantees the resilience of LSTM. Specifically, even if temporary perturbations in the input sequence disrupt inference, the state variables can converge to the same inference values that would have been obtained without perturbations. Sufficient conditions for the LSTM to exhibit the $\delta$ISS properties have already been derived as inequalities of parameters (Terzi et al., 2021; Bonassi et al., 2023). However, it is not clarified in these studies how long it takes for state variables to converge. We will refine and improve upon these previous studies to achieve the above three goals.

### 1.3 Contribution

The main contributions of this study are as follows:

**Theoretical Contributions**

- Refinement of an evaluation of LSTM's invariant sets, which form the foundation for stability analysis (Section 4).

- Improvement of sufficient conditions for LSTM to be $\delta$ISS (Section 4).

- Definition of recovery time that quantifies the resilience of LSTM and derivation of a data-independent upper bound for it (Section 5).

**Practical Contributions**

- Introduction of a quantitative metric for quality assurance through the recovery time (Section 6).

- Development of a method to control the trade-off between recovery time and inference accuracy for LSTMs (Section 6).

To verify the above contributions, simple experiments are conducted in Section 7. Furthermore, we compared our method with conventional data-driven methods in Section 8.

## 2 Related Work

We describe the related study on the control theory, theoretical LSTM stability, and experimental LSTM quality.

### 2.1 Control System and Stability Theory

In control theory, stability and resilience are important properties. Furthermore, it is well-established that the stability of controlled objects fundamentally determines the overall stability and resilience of control systems. For instance, systems possessing input-to-state stability (ISS) properties achieve global asymptotic stability under closed-loop control (Jiang & Wang, 2001). Moreover, systems with $\delta$ISS characteristics acquire robustness when implemented with model predictive control (MPC) (Bayer et al., 2013).

Therefore, in control methods such as MPC where controlled objects are modeled as dynamical systems, stability requirements for the dynamical systems are essential. While stability can be comprehensively guaranteed by general theory for linear systems, practical applications require specific and detailed engineering for a nonlinear system. This research focuses on LSTM networks as a specific case of nonlinear systems.

### 2.2 Theoretical LSTM Stability Assurance

Theoretical approaches have investigated sufficient conditions for LSTM networks to satisfy various stability properties. For single-layer LSTMs, conditions for global asymptotic stability (Deka et al., 2019), ISS (Bonassi et al., 2020), and $\delta$ISS (Terzi et al., 2021) have been successively established. Research on sufficient conditions for $\delta$ISS has been extended to multilayer LSTM architectures (Bonassi et al., 2023).

However, existing studies face a significant limitation: while they can guarantee recovery, they cannot provide estimates of time to recover. Our research addresses these challenges by advancing the stability theory of LSTM to enable both quantitative evaluation and adjustment of LSTM resilience.

### 2.3 Experimental LSTM Quality Assurance

This paper discusses theoretical LSTM quality assurance; however, there are also some experimental methods. For example, Wehbe et al. (2018) evaluated the robustness of LSTM inference using noise-added datasets. Ahmed et al. (2020) proposed creating robust LSTMs by expanding training datasets to include abnormal scenes. Grande et al. (2021) proposed a data-driven stability assurance method.

These data-driven verification methods, like the ones mentioned above, are comprehensive when sufficient data are available, yet face challenges in determining adequate data quantities and provide only sparse guarantees for nonlinear models like LSTMs. Theoretical verification offers limited but exhaustive guarantees within its scope, and is therefore complementary to, rather than in conflict with, a data-driven approach.

## 3 Preliminary

In this section, we introduce essential mathematical preliminaries, an important stability concept: $\delta$ISS, and the LSTM.

### 3.1 Notation

For a vector $v \in \mathbb{R}^n$, $\|v\| = \|v\|_2$ represents the Euclidean norm of $v$, and $\|v\|_\infty = \max_{i \in [n]} |v_{(i)}|$ represents the maximum norm, where $v_{(i)}$ is the $i$-th component of $v$ and $[n] = \{1, 2, \ldots, n\}$. $v_+$ denotes a vector $(\max\{v_{(i)}, 0\})_{i \in [n_c]}$ for a vector $v$, and $\mathbf{1}_n$ is a $n$-dimensional vector of all components being one. For a discrete-time sequence of vector $v(k), k \in \mathbb{Z}_{\geqslant 0}$, we denote $v(k_1 : k_2) = [v(k_1), \ldots, v(k_2)]$, and $\|v(k_1 : k_2)\|_{2,\infty} = \|(\|v(k_1)\|_2, \ldots, \|v(k_2)\|_2)^\top\|_\infty$.

For a matrix $A \in \mathbb{R}^{m \times n}$, $\|A\| = \|A\|_2$ represents the induced 2-norm of $A$, and $\|A\|_\infty$ does the induced $\infty$-norm, that is, $\|A\|_\infty = \max_i \sum_j |A_{(i,j)}|$, where $A_{(i,j)}$ is the $(i,j)$ entry of $A$. In addition, $|A|$ denotes a matrix whose components are the absolute values of $A$, that is, $|A| = (|A_{(i,j)}|)$. For a matrix $A \in \mathbb{R}^{n \times n}$, $\rho(A)$ denotes the spectral radius of $A$, that is, $\rho(A) = \max_{i \in [n]} |\lambda_i|$, where $\lambda_i$ are the eigenvalues of $A$.

We denote the sigmoid function as $\sigma(w) = \frac{1}{1+e^{-w}}$ and the hyperbolic tangent as $\phi(w) = \tanh w = \frac{e^w - e^{-w}}{e^w + e^{-w}}$.

### 3.2 Stability

**Definition 3.1** (Class $\mathcal{K}$, $\mathcal{KL}$, and $\mathcal{K}_\infty$). A function $\alpha : \mathbb{R}_{\geqslant 0} \to \mathbb{R}_{\geqslant 0}$ is called $\mathcal{K}$-function if it is continuous, strictly increasing and $\alpha(0) = 0$. A function $\beta : \mathbb{R}_{\geqslant 0} \times \mathbb{R}_{\geqslant 0} \to \mathbb{R}_{\geqslant 0}$ is called $\mathcal{KL}$-function if $\beta(\cdot, t)$ is $\mathcal{K}$-function for each $t \geqslant 0$, and $\beta(s, \cdot)$ is decreasing and $\beta(s, t) \to 0$ as $t \to \infty$ for each $s \geqslant 0$. A function $\gamma : \mathbb{R}_{\geqslant 0} \to \mathbb{R}_{\geqslant 0}$ is called $\mathcal{K}_\infty$-function if it is $\mathcal{K}$-function and $\lim_{t \to \infty} \gamma(t) = \infty$.

Now we consider a (autonomous) discrete-time nonlinear system

$$s(t+1) = f(s(t), x(t)) \tag{1}$$

for initial state $s(0)$ and input $x(t), t \geqslant 0$. In the following, we denote $s_i(t)$ as the system whose initial state is $s_i(0)$ and input is $x_i(t)$.

**Definition 3.2** ($\delta$ISS, Angeli 2002; Bayer et al. 2013). System (1) is called incrementally input-to-state stable ($\delta$ISS) if there exist continuous functions $\beta \in \mathcal{KL}$ and $\gamma \in \mathcal{K}_\infty$ such that, for any initial states $s_1(0), s_2(0) \in \mathbb{R}^{n_s}$ and for any input sequence $x_1(0 : t), x_2(0 : t) \in [-x_{\max}, x_{\max}]^{n_x \times (t+1)}$, $s_1(t)$ and $s_2(t)$ satisfy

$$\|s_1(t) - s_2(t)\| \leqslant \beta(\|s_1(0) - s_2(0)\|, t) + \gamma(\|x_1(0 : t) - x_2(0 : t)\|_{2,\infty})$$

for all $t \in \mathbb{Z}_{\geqslant 0}$ and input domain radius $x_{\max} > 0$.

### 3.3 LSTM Network

The LSTM is described by the following relations:

$$c^{(l)}(t+1) = \sigma(W_f^{(l)} x^{(l)}(t) + U_f^{(l)} h^{(l)}(t) + b_f^{(l)}) \odot c^{(l)}(t)$$
$$+ \sigma(W_i^{(l)} x^{(l)}(t) + U_i^{(l)} h^{(l)}(t) + b_i^{(l)}) \odot \phi(W_c^{(l)} x^{(l)}(t) + U_c^{(l)} h^{(l)}(t) + b_c^{(l)}), \tag{2a}$$

$$h^{(l)}(t+1) = \sigma(W_o^{(l)} x^{(l)}(t) + U_o^{(l)} h^{(l)}(t) + b_o^{(l)}) \odot \phi(c^{(l)}(t+1)), \tag{2b}$$

$$y(t) = U_y h^{(L)}(t) + b_y, \tag{2c}$$

$$x^{(l)}(t) = \begin{cases} x & \text{for } l = 1, \\ h^{(l-1)}(t+1) & \text{for } l = 2, \ldots L, \end{cases} \tag{2d}$$

where $x \in [-x_{\max}, x_{\max}]^{n_x}$, $c^{(l)} \in \mathbb{R}^{n_c}$, $h^{(l)} \in (-1, 1)^{n_c}$, and $y \in \mathbb{R}^{n_y}$ are the input, the cell state, the hidden state, and the output, respectively, for each time $t \in \mathbb{Z}_{\geqslant 0}$, each layer $l \in [L]$, and $\odot$ denotes the element-wise product. Note that $x_{\max} = 1$ if $l \geqslant 2$. Moreover, $W_*^{(l)} \in \mathbb{R}^{n_c \times n_x}, U_*^{(l)} \in \mathbb{R}^{n_c \times n_c}, U_y \in \mathbb{R}^{n_y \times n_c}$ are the weight matrices, $b_*^{(l)} \in \mathbb{R}^{n_c}$ and $b_y \in \mathbb{R}^{n_y}$ are the biases, where $*$ denotes $f, i, c$, or $o$.

In this paper, we regard the LSTM as a discrete-time nonlinear system (1) by defining the state $s(t)$ of (1) as $s^{(1:L)}(t) := [s^{(1)}(t), \ldots, s^{(L)}(t)]$, where $s^{(l)}(t) := (c^{(l)}(t)^\top, h^{(l)}(t)^\top)^\top$.

# 4 Improvement of Incremental ISS Condition

In this section, we relax the existing $\delta$ISS condition of the LSTM (Terzi et al., 2021) by refining the invariant set of the LSTM dynamics. This relaxation contributes to an accurate evaluation of the recovery time introduced in Section 5.

## 4.1 Invariant Set in LSTM

Define $G_*^{(l)}, G_c^{(l)} : \mathbb{R}_{\geqslant 0} \to \mathbb{R}_{\geqslant 0}$ ($*$ denotes $f, i$, or $o$) by

$$G_*^{(l)}(\eta) = \left\| \left( x_{\max} |W_*^{(l)}| \mathbf{1}_{n_x} + \eta |U_*^{(l)}| \mathbf{1}_{n_c} + b_*^{(l)} \right)_+ \right\|_\infty,$$

$$G_c^{(l)}(\eta) = \left\| x_{\max} |W_c^{(l)}| \mathbf{1}_{n_x} + \eta |U_c^{(l)}| \mathbf{1}_{n_c} + |b_c^{(l)}| \right\|_\infty.$$

In addition, we define the sequences $\{\overline{\sigma}_*^{(l)}(k)\}_{k=0}^\infty$, $\{\overline{\phi}_c^{(l)}(k)\}_{k=0}^\infty$, $\{\eta^{(l)}(k)\}_{k=-1}^\infty$, and $\{\overline{c}^{(l)}(k)\}_{k=0}^\infty$ using following recursive formulas:

$$\overline{\sigma}_*^{(l)}(k) = \sigma(G_*^{(l)}(\eta^{(l)}(k-1))), \tag{3}$$

$$\overline{\phi}_c^{(l)}(k) = \phi(G_c^{(l)}(\eta^{(l)}(k-1))), \tag{4}$$

$$\eta^{(l)}(k) = \phi(\overline{c}^{(l)}(k))\overline{\sigma}_o^{(l)}(k), \tag{5}$$

$$\overline{c}^{(l)}(k) = \frac{\overline{\sigma}_i^{(l)}(k)\overline{\phi}_c^{(l)}(k)}{1 - \overline{\sigma}_f^{(l)}(k)},$$

for $k \in \mathbb{Z}_{\geqslant 0}$, with initial term $\eta^{(l)}(-1) = 1$, where $*$ denotes $f, i$ or $o$. Moreover, we define that

$$\mathcal{C}^{(l)}(k) := \left\{ c^{(l)} \in \mathbb{R}^{n_c} : \|c^{(l)}\|_\infty \leqslant \overline{c}^{(l)}(k) \right\},$$

$$\mathcal{H}^{(l)}(k) := \left\{ h^{(l)} \in \mathbb{R}^{n_c} : \|h^{(l)}\|_\infty \leqslant \phi(\overline{c}^{(l)}(k))\overline{\sigma}_o^{(l)}(k) \right\}.$$

**Proposition 4.1** (Invariant Set). *Let $L \in \mathbb{N}$ and $k \in \mathbb{Z}_{\geqslant 0}$. $\mathcal{S}^{(l)}(k) := \mathcal{C}^{(l)}(k) \times \mathcal{H}^{(l)}(k)$ is an invariant set, that is, for all $t \in \mathbb{Z}_{\geqslant 0}$, $l \in [L]$ and $x \in [-x_{\max}, x_{\max}]^{n_x}$, if $s^{(l)}(0) = (c^{(l)}(0)^\top, h^{(l)}(0)^\top)^\top \in \mathcal{S}^{(l)}(k)$, then $s^{(l)}(t) = (c^{(l)}(t)^\top, h^{(l)}(t)^\top)^\top \in \mathcal{S}^{(l)}(k)$. Furthermore, $\{\mathcal{S}^{(l)}(k)\}_{k=0}^\infty$ is a decreasing sequence in the sense of set inclusion.*

*Proof sketch.* Since the activation functions are strictly monotonically increasing and bounded, the sequences are monotonically decreasing and converge to some limit. This proposition is proved by induction on $k$. □

The exact proof is described in Appendix A.1. We note that the invariant set $\mathcal{S}^{(l)}(0)$ is considered in the literature (Terzi et al., 2021; Bonassi et al., 2023).

## 4.2 Incremental ISS of LSTM

Using Proposition 4.1, we provide the following sufficient condition of $\delta$ISS:

**Theorem 4.2** ($\delta$ISS of LSTM). *Let $L \in \mathbb{N}$ and $k \in \mathbb{Z}_{\geqslant 0}$. We assume that $s^{(l)}(0) \in \mathcal{S}^{(l)}(k)$ for all $l \in [L]$. The LSTM is $\delta$ISS if $\rho(A_s^{(l)}(k)) < 1$ for all $l \in [L]$, where*

$$A_s^{(l)}(k) = \begin{bmatrix} \overline{\sigma}_f^{(l)}(k) & \alpha_s^{(l)}(k) \\ \overline{\sigma}_o^{(l)}(k)\overline{\sigma}_f^{(l)}(k) & \alpha_s^{(l)}(k)\overline{\sigma}_o^{(l)}(k) + \frac{1}{4}\phi(\overline{c}^{(l)}(k))\|U_o^{(l)}\| \end{bmatrix},$$

$$\alpha_s^{(l)}(k) = \frac{1}{4}\|U_f^{(l)}\|\overline{c}^{(l)}(k) + \overline{\sigma}_i^{(l)}(k)\|U_c^{(l)}\| + \frac{1}{4}\|U_i^{(l)}\|\overline{\phi}_c^{(l)}(k).$$

*Furthermore, $\rho(A_s^{(l)}(k))$ is monotonically decreasing with respect to $k \in \mathbb{Z}_{\geqslant 0}$.*

*Proof sketch.* From the invariant set in Proposition 4.1 and Lipschitz continuity of the activation functions, we derive a matrix $A_s^{(l)}(k)$ and a constant $a_x^{(l)}(k)$ (as given in Proposition 5.3) such that the following inequality holds:

$$\begin{pmatrix} \left\| c_1^{(l)}(t+1) - c_2^{(l)}(t+1) \right\| \\ \left\| h_1^{(l)}(t+1) - h_2^{(l)}(t+1) \right\| \end{pmatrix} \leqslant A_s^{(l)}(k) \begin{pmatrix} \left\| c_1^{(l)}(t) - c_2^{(l)}(t) \right\| \\ \left\| h_1^{(l)}(t) - h_2^{(l)}(t) \right\| \end{pmatrix} + a_x^{(l)}(k) \left\| x_1^{(l)}(t) - x_2^{(l)}(t) \right\|.$$

The monotonic decrease of $\rho(A_s^{(l)}(k))$ is demonstrated using the Perron–Frobenius theorem. □

The rigorous proof is described in Appendix A.2, and where we also see that a similar relaxation can be obtained for ISS.

*Remark* 4.3. We give some important observations to Theorem 4.2:

(i) Theorem 4.2 coincides with existing results (Terzi et al., 2021; Bonassi et al., 2023) for the case $k = 0$. Monotonically decreasing $\rho(A_s^{(l)}(k))$ makes it possible to relax the constraints on the model in the case of $k > 0$.

(ii) Because $\mathcal{S}^{(l)}(k)$ is a contracting sequence, the assumption for the initial state becomes a stronger constraint as $k$ becomes larger. However, since the initial states of LSTMs are usually set to zero in practice, we believe that there is no practical limitation.

(iii) From the above reasons, it is desirable to set $k$ to infinity. However, because it was difficult to find the limit, we need to calculate the recurrence formula in Section 4.1. This is required every time we update the parameters in the training, so the amount of calculation increases with $k$.

## 5 Recovery Time

In this section, we define a novel mathematical formulation, the recovery time of LSTM, which accurately expresses the time required for the system to recover from disturbances and return to its normal state. This definition makes it difficult to evaluate in practical applications due to data dependence, but we overcome this limitation by deriving a practical and computable upper bound for the recovery time. This upper bound constitutes the primary novel contribution of our research, offering a fresh perspective on LSTM performance evaluation.

**Definition 5.1** (Recovery Time). Let inputs $x(t), \widehat{x}(t) \in [-x_{\max}, x_{\max}]^{n_x}$ for $t \in \mathbb{Z}_{\geqslant 0}$. Assume that there exists $t_0 > 0$ such that $x(t) = \widehat{x}(t)$ for any $t \geqslant t_0$. We define *recovery time* $T_R \geqslant 0$ for given $e \geqslant 0$ as

$$T_R(x, \widehat{x}, s(0); e) = \min\{t \geqslant t_0 : \|y(t', s(0), x(0:t')) - y(t', s(0), \widehat{x}(0:t'))\| \leqslant e \text{ for any } t' \geqslant t\} - t_0,$$

and as $T_R = \infty$ if the set on the right-hand side is empty.

*Remark* 5.2. Note that the recovery time $T_R$, depending on the input data, can be defined similarly for any time-series model. While the specific formulas of the upper bound for $T_R$ (given in Theorem 5.4) differ across models, the underlying concept remains consistent.

Evaluating the recovery time $T_R$ is difficult because it depends on the input data, so we utilize the function $\beta$ of $\delta$ISS (Definition 3.2): Theorem 4.2 indicates that there exists a $\mathcal{KL}$ function $\beta$ satisfying

$$\|s(t, s_1(0), x(0:t)) - s(t, s_2(0), x(0:t))\| \leqslant \beta(\|s_1(0) - s_2(0)\|, t) \tag{6}$$

for any initial states $s_1(0), s_2(0) \in \prod_{l=1}^{L} \mathcal{S}^{(l)}(k)$ and input sequence $x(0:t) \in [-x_{\max}, x_{\max}]^{n_x \times (t+1)}$ of the LSTM and $t > 0$. The proof of Theorem 4.2 also leads to a more precise and explicit evaluation of (6):

**Proposition 5.3** (Explicit Representation of $\delta$ISS). *For* $\zeta^{(1:L)} := (\zeta^{(1)}, \ldots, \zeta^{(L)}) \in \mathbb{R}_+^{1 \times L}$, *let*

$$\tilde{\beta}(\zeta^{(1:L)}, t; k) = \mu^{(L)}(t)\rho(A_s^{(L)}(k))^t \zeta^{(L)}$$
$$+ \sum_{l=1}^{L-1} \mu^{(l:L)}(t) \frac{(t+L-l)!}{t!(L-l)!} \max_{i=l,\cdots,L} \rho(A_s^{(i)}(k))^t \left( \prod_{i=l+1}^{L} \|a_x^{(i)}(k)\| \right) \zeta^{(l)}.$$

*Then, it satisfies for any initial states $s_1(0), s_2(0) \in \prod_{l=1}^{L} \mathcal{S}^{(l)}(k)$ and any input sequence $x(0:t) \in [-x_{\max}, x_{\max}]^{n_x \times (t+1)}$ of the LSTM and for $t > 0$ that*

$$\|s^{(L)}(t, s_1(0), x(0:t)) - s^{(L)}(t, s_2(0), x(0:t))\| \leqslant \tilde{\beta}(\|s_1^{(1)}(0) - s_2^{(1)}(0)\|, \dots, \|s_1^{(L)}(0) - s_2^{(L)}(0)\|, t; k).$$

*Here,*

$$a_x^{(l)}(k) = \begin{pmatrix} \alpha_x^{(l)}(k) \\ \alpha_x^{(l)}(k)\bar{\sigma}_o^{(l)}(k) + \frac{1}{4}\phi(\bar{c}^{(l)}(k))\|W_o^{(l)}\| \end{pmatrix},$$

$$\alpha_x^{(l)}(k) = \frac{1}{4}\|W_f^{(l)}\|\bar{c}^{(l)}(k) + \bar{\sigma}_i^{(l)}(k)\|W_c^{(l)}\| + \frac{1}{4}\|W_i^{(l)}\|\bar{\phi}_c^{(l)}(k),$$

$$\mu^{(l:L)}(t) = \prod_{i=l}^{L} \mu^{(i)}(t), \mu^{(l)}(t) = \sqrt{(1 + (r^{(l)})^{2t}) + \frac{|\nu^{(l)}|^2}{|\lambda_1^{(l)}|^2}\left(\frac{1 - (r^{(l)})^t}{1 - r^{(l)}}\right)^2},$$

*for any decomposition such as*

$$A_s^{(l)}(k) = U \begin{bmatrix} \lambda_1^{(l)} & \nu^{(l)} \\ 0 & \lambda_2^{(l)} \end{bmatrix} U^*, \lambda_1^{(l)} \geqslant \lambda_2^{(l)}, r^{(l)} := |\lambda_2^{(l)}|/|\lambda_1^{(l)}| \in [0, 1],$$

*where $U$ is a suitable unitary matrix (note that any matrix is triangulizable by some regular matrix $U$). Also, we regard the term of $\sum_{l=1}^{0}$ as 0.*

*Proof sketch.* This representation is given in the proof of Theorem 4.2, specifically by repeating the evaluation of the inputs of each layer with the state variables of front layers according to (2d). □

**Theorem 5.4** (Upper Bound of Recovery Time). *We assume for all $l \in [L]$ that $s^{(l)}(0) \in \mathcal{S}^{(l)}(k)$ and $\rho(A_s^{(l)}(k)) < 1$. Let infinite length inputs $x(t), \hat{x}(t) \in [-x_{\max}, x_{\max}]^{n_x}, t \geqslant 0$ and $e > 0$. We assume that there exists $t_0$ such that $x(t) = \hat{x}(t)$ for any $t \geqslant t_0$. Then, we have*

$$\sup_{s(0) \in \prod_{l=1}^{L} S_k^{(l)}} T_R(x, \hat{x}, s(0); e) \leqslant \min\{t \geqslant 0 : \tilde{\beta}(2\sqrt{n_c}\bar{s}(k), t'; k) \leqslant e/\|U_y\| \text{ for any } t' \geqslant t\},$$

*where $\bar{s}(k) = (\bar{s}^{(1)}(k), \dots, \bar{s}^{(L)}(k))$ and $\bar{s}^{(l)}(k) = \|(\bar{c}^{(l)}(k), \phi(\bar{c}^{(l)}(k))\bar{\sigma}_o^{(l)}(k))\|$.*

*Proof sketch.* This is from Definition 5.1, Proposition 5.3 (shifting the initial time by $t_0$), and the fact that the diameter of the invariant set is $2\sqrt{n_c}\bar{s}^{(l)}(k)$. □

Detailed proof is stated in Appendix A.3.

*Remark* 5.5. Proposition 5.3 and Theorem 5.4 imply the following:

(i) $\tilde{\beta}(\zeta^{(1:L)}, t; k)$ is calculated from the weight of LSTM and the recursive formulas (3), (4), and (5). Since recovery time can be estimated using only $\tilde{\beta}(2\sqrt{n_c}\bar{s}(k), t; k)$, this upper bound is a data-independent formula.

(ii) As $k$ increases, $\rho(A_s^{(l)}(k))$ decreases, consequently leading to a reduction in $\tilde{\beta}(2\sqrt{n_c}\bar{s}(k), t; k)$. This reduction in $\tilde{\beta}$ results in a sharper upper bound for $T_R$.

# 6 Resilience-Aware Training and Evaluation Methods

We provide a comprehensive strategy for translating theoretical foundations into practical applications.

## 6.1 Estimate

**Aim**   Quantitatively assess the resilience of LSTMs.

**Method**  Evaluate the obtained model's resilience using the following index:

$$\bar{T}_R(e;k) := \min\{t \geqslant 0 : \tilde{\beta}(2\sqrt{n_c}\bar{s}(k), t'; k) \leqslant e/\|U_y\| \text{ for any } t' \geqslant t\}$$

for some $k$. Since $\bar{T}_R$ depends solely on the model's weight parameters, it can be evaluated independently of the data. According to Theorem 5.4, a smaller $\bar{T}_R$ indicates a stronger resilience capacity of the model.

**Use case**  We propose to establish this metric as a formal requirement for assessing LSTM quality, contextualized within the framework of control periods.

## 6.2 Training

**Aim**  Build an LSTM with a specified degree of resilience.

**Method**  Control the model's resilience (i.e., $T_R$) during training by the following loss function:

$$Loss = \mathcal{L} + \lambda\Phi(\varepsilon), \tag{7}$$

$$\Phi(\varepsilon) = \sum_{l=1}^{L} \max\{\rho(A_s^{(l)}(k)) - 1 + \varepsilon, 0\}, \tag{8}$$

where $\mathcal{L}$ denotes the existing loss function, $\Phi$ denotes the penalty term for the resilience, $\lambda > 0$ denotes the penalty intensity, and $\varepsilon \in [0, 0.5]$ denotes the control parameter for the resilience. The resilience of LSTM can be ensured through the design of the loss function, without modifications to the model architecture. We can't directly penalize $T_R$ or $\bar{T}_R$ because these are initially unbounded, which prevents parameter updates. The reason we penalize $\rho(A_s^{(l)}(k))$ is that the recovery time can be approximated as $\bar{T}_R \approx (-\log\rho(A_s^{(l)}(k)))^{-1}$ by focusing on the exponential term, which dominates decay.

The resilience of LSTM can be precisely calibrated by adjusting the parameter $\varepsilon$. In fact, as training progresses, $\Phi(\varepsilon)$ is expected to approach 0, and after training is completed, the model is expected to satisfy $\rho(A_s^{(l)}(k)) \approx 1 - \varepsilon$. Consequently, increasing the value of $\varepsilon$ enhances the resilience of the model. This amplification of resilience concurrently imposes stronger constraints on the model, potentially leading to a trade-off between resilience and model accuracy.

**Use case**  We tune $\varepsilon$ to balance resilience and accuracy requirements, observing the trade-off and selecting an optimal model.

## 7  Experiments

In this section, we verify the following two properties of our proposed method:

(A) Increasing $k$ in $\bar{T}_R(k)$ leads to more accurate recovery time estimates.

(B) Adjusting $\varepsilon$ enables the tuning of $T_R$, and there is a trade-off between recovery time and inference loss.

Detailed experimental settings are given in Appendix C.

## 7.1 Experiment I: Simplified Model

In this section, we employ a simplified model to validate (A).

### 7.1.1 Experimental Setup

We employed a simplified model using an LSTM network with randomly initialized weight parameters. We applied step input to this model, introduced instantaneous noise, and observed the time required for inference recovery (for a visual representation of the recovery time measurement, see Figure 1).

### 7.1.2 Results and Analysis

We calculated $T_R$ and $\bar{T}_R(k)$ ($k = 0, 1, \cdots, 20$) for 20 randomly initialized models, deriving estimated average test errors and correlation coefficients. As shown in Figure 2, the estimation test error consistently decreases as $k$ increases, eventually converging at around $k = 20$ steps. Furthermore, the correlation coefficient exhibits a generally increasing (except for a temporary decrease) trend as $k$ increases, which indicates a strong relationship between the true and our estimated values.

Figure 3 compares the estimation accuracy for $k = 0$ and $k = 20$, demonstrating improved accuracy with a larger $k$. The left side of the red line is the area where $\delta$ISS is guaranteed by Theorem 4.2. Increasing $k$ from 0 to 20 decreases $\rho(A(k))$ by 32% on average, then all data points are included in the $\delta$ISS guaranteed area.

To illustrate this improvement in a specific case, we focus on one of these models. As evident in Figure 4, the $\bar{T}_R(k)$ value for $k = 20$ is smaller than that for $k = 0$. Furthermore, this value is closer to the true $T_R$. This indicates that using a larger $k$ value enables a more accurate assessment.

These results offer empirical support for our theoretical statements in Theorem 5.4 and Remark 5.5: increasing $k$ not only relaxes the sufficient condition of $\delta$ISS but also enhances the accuracy of recovery time estimation.

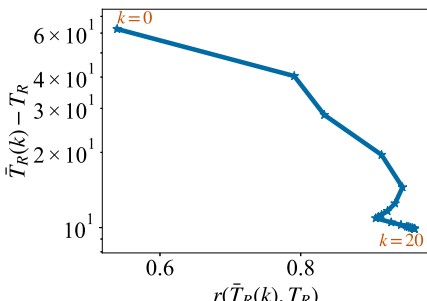

Figure 2: k-dependency of $\bar{T}_R(k)$. Each point represents a pair of the correlation coefficient and estimation error between $\bar{T}_R(k)$ and $T_R$ for $k = 0, 1, \ldots, 20$ (averaged over 20 models). Edges connect points corresponding to consecutive $k$ values. The figure shows that, as $k$ increases, the correlation coefficient tends to increase, and the estimation error monotonically decreases.

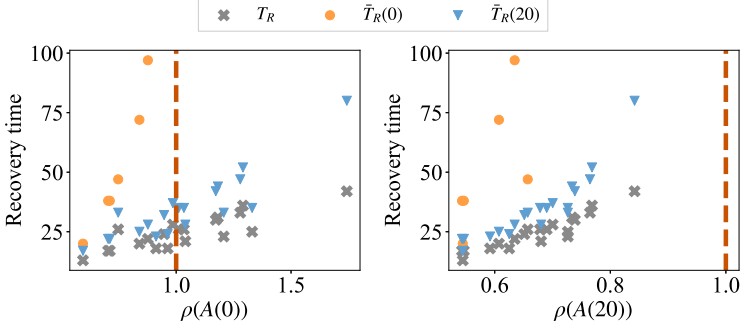

Figure 3: Accuracy of recovery time estimation. Gray dots represent $T_R$ for each model. Orange and blue dots indicate $\bar{T}_R(k)$ in $k = 0$ and $k = 20$ for each model. The red line represents the $\delta$ISS sufficient condition boundary. The models in the left side area of the red line are guaranteed $\delta$ISS by Theorem 4.2. Increasing $k$ from 0 to 20 decreases $\rho(A(k))$ by 32% on average over 20 models. This figure indicates that increasing $k$ guarantees more models to be $\delta$ISS.

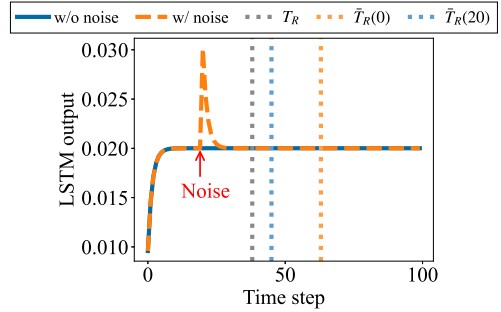

Figure 4: Recovery time of simplified model. The blue and orange lines depict the LSTM outputs for the normal input without noise and the anomalous input subjected to instantaneous noise, respectively. Vertical lines denote the recovery times, indicating that using a larger $k$ value enables a more accurate assessment.

## 7.2 Experiment II: Two-Tank System

Here, we demonstrate the effectiveness of our approach from the perspective of (B) in a more practical scenario using the two-tank system, a recognized benchmark in control experiments (Jung et al., 2023; Schoukens & Noël, 2017).

### 7.2.1 Experimental Setup

**System**   The two-tank system is described by the following differential equations (Schoukens & Noël, 2017):

$$\frac{dh_1}{dt} = -p_1\sqrt{2gh_1} + p_2 u,$$
$$\frac{dh_2}{dt} = p_3\sqrt{2gh_1} - p_4\sqrt{2gh_2},$$

where $u$ is the control input, $h_1$ and $h_2$ are the tank levels, $g$ is the gravitational acceleration and $p_1$, $p_2$, $p_3$, and $p_4$ are hyperparameters depending on the system properties. The LSTM model is designed to predict the water level at the next time step based on the current water level and control input. The current water level is obtained from sensors, which are assumed to be subject to sensor noise. Formally, we define the input-output relationship of the LSTM as $x(t) = (u(t), h(t) + w(t))$ and $y(t) = h(t+1)$, where $w \sim \mathcal{N}(0, 0.01^2)$.

**Baseline**   To evaluate our proposed recovery time estimation method and adjustment technique, we prepared a baseline model: a standard model trained without considering resilience. As shown in Figure 5, the baseline model closely approximates the target system for inference.

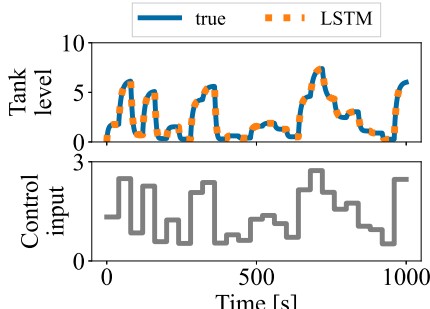

Figure 5: Baseline of two-tank system. Top: The blue line depicts the resulting tank water level $h_1$ while the orange line illustrates the LSTM prediction. Bottom: control input signal. We can confirm that the baseline model closely approximates the target system.

## 8 Theoretical Approaches Compare to Data-Driven Approaches

We conducted comparative experiments between our proposed theoretical method and conventional data-driven approaches. In the data-driven approach, training was performed by randomly adding pulses along the time series direction to the training data.

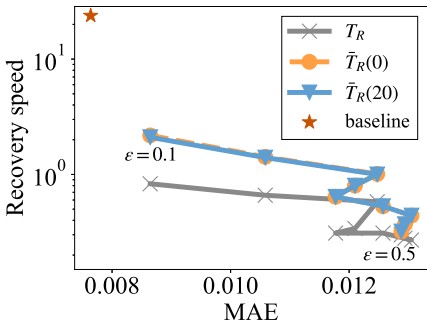

Figure 6: Trade-off controlled by $\varepsilon$. The red dot represents the baseline value. For each $\varepsilon$ from 0.1 to 0.5 in 0.05 increments value, gray, orange, and blue dots represent MAE and $T_R, \bar{T}_R(0)$, and $\bar{T}_R(20)$ pairs, respectively, where MAE is the loss calculated on test data. The figure shows a trade-off between recovery times and MAE when comparing the baseline and our models, and when varying $\varepsilon$ of our model.

## 8.1 Setting

### 8.1.1 TRAIN

**Data set**   For the adversarial learning, training and validation data were generated by adding pulse signals to the input data generated in Section C.2, excluding control inputs. For data with time series length $T = 50,000$, we generated randomly occurring sustained pulses. Each pulse occurrence follows a Poisson process with an average rate of $\lambda = 0.001$ per unit time. The total number of pulses over the entire period $T$ is determined according to the Poisson distribution $\text{Poisson}(\lambda T)$.Each pulse has the following characteristics:

- The start time is randomly selected from a uniform distribution in the range $[0, T - d]$, where $d = 10$ is the duration of each pulse.

- The amplitude is sampled from a uniform distribution within $[0, M]$, where $M$ is the maximum intensity of the pulse.

- Each pulse is added to the input signal at a constant value for $d$ steps from the selected start time.

We created training and validation data with variations of $M = 1, 3, 10$ using the method described above. Models trained with each of these datasets are referred to as "DD pulse $M$ model" (where DD stands for data-driven), while our model was trained using data without added pulses.

**Loss**   We used the same loss function as defined in Section C.2.

**Hyperparameters**   With the exception of the following two points, we used the same hyperparameters as in Section C.2:

- For our model, $\varepsilon = 0.05$.

- For the DD pulse $M$ models, $\lambda = 0$.

### 8.1.2 EVALUATION

We evaluated each model using the following evaluation data, under the same settings as in Section C.2.

We define $\hat{x}_p$ that satisfies the assumptions of Theorem 5.4 with $t_0 = 1010$ as

$$\hat{x}_p(t) = \begin{cases} (u(t), h_1(t) + p, h_2(t) + p) & \text{if } 1000 \leqslant t \leqslant 1010, \\ x(t) & \text{else,} \end{cases}$$

where $p$ was selected from the set $\{1.0, 5.0, 9.0\}$, and let $\widehat{y}_p$ represent the output of the LSTM when $\widehat{x}_p$ is used as the input.

## 8.2 Results and Analysis

As shown in Table 1, regarding $T_R$, our proposed method demonstrated superiority. Our model achieved faster recovery times compared to models obtained using data-driven methods. Although models trained with data having stronger noise pulses showed improvement in recovery time, they could not reach the performance of our theoretically guaranteed model. These results indicate that our proposed method is effective in enhancing LSTM's recovery capability.

In achieving stability criteria, our model with theoretical guarantees also showed clear advantages. Models trained with these data-driven techniques failed to meet the theoretical stability criterion of $\rho(A_s) < 1$. This result suggests that approaches relying solely on data are insufficient for theoretical guarantees of system resilience.

On the other hand, in terms of noise resistance, models trained with data-driven approaches outperformed our model in certain cases. This is likely because our proposed method did not directly aim to maximize noise resistance. Noise resistance can be theoretically bounded using $\gamma$ from Definition 3.2. As a future challenge, by explicitly deriving $\gamma$, there is potential to improve noise resistance metrics.

In general, our theoretical approach has been proven to provide a better solution than data-driven methods to ensure resilience under any noise conditions.

|  | Baseline | DD pulse 1 | DD pulse 5 | DD pulse 10 | Our |
|---|---|---|---|---|---|
| MAE | **0.007** | 0.017 | 0.014 | 0.012 | 0.010 |
| $T_R(x, \widehat{x}_1)$ | 21.31 | 17.04 | 9.31 | 4.49 | **0.86** |
| $T_R(x, \widehat{x}_5)$ | 22.91 | 19.74 | 8.11 | 4.74 | **1.04** |
| $T_R(x, \widehat{x}_9)$ | 24.39 | 17.33 | 8.11 | 4.90 | **1.04** |
| $\overline{T}_R$ | $\infty$ | $\infty$ | $\infty$ | $\infty$ | **19.32** |
| $\rho(A)$ | 1.537e+11 | 6.001e+7 | 2.627e+12 | 1.295e+6 | **0.9476** |
| $\max_t |y(t) - \widehat{y}_1(t)|$ | **0.03** | 0.06 | 0.16 | 0.26 | 0.21 |
| $\max_t |y(t) - \widehat{y}_5(t)|$ | 0.40 | **0.35** | 1.83 | 3.05 | 1.30 |
| $\max_t |y(t) - \widehat{y}_9(t)|$ | 1.23 | **0.82** | 3.30 | 5.88 | 2.80 |

Table 1: Comparison of stability performance between theoretical approaches and data-driven approaches. "DD pulse $M$" refers to a model trained with random pulses added to the training data, where $M$ indicates the upper limit of the pulse magnitude. "Our" refers to a model trained using our proposed loss function (7). Bold numbers indicate the best performance for each metric.

### 8.2.1 Results and Analysis

We will compare the models trained using our proposed loss function (7) against the baseline model. First, regarding (A), although the result in Figure 6 does not contradict Theorem 5.4 in the sense of weakly decreasing, the magnitude of this decay was notably smaller than the results in Section 5. Next, concerning (B), we observed a trade-off between recovery times and MAE loss when comparing the baseline model and our models, and when varying $\varepsilon$ of our model, as illustrated in Figure 6. Specifically, as $\varepsilon$ increases from 0.1 to 0.5, we observe a trend of increasing MAE loss and decreasing recovery time. This trend aligns with our theoretical predictions in Section 6.2. The experimental results demonstrate that LSTM resilience can be tuned via $\varepsilon$, validating our proposed method (8) that introduces recovery time as a quality assurance metric and balances inference accuracy with recovery time.

## 9 Conclusion and Limitation

In this paper, we introduced and theoretically estimated the recovery time. Furthermore, we established a practical framework for resilience-aware training and evaluation of recovery time. However, there are some limitations in our research:

1. Theoretical and data-independent guarantees do not always reflect actual data distributions. Therefore, we strongly recommend complementing our approach with data-driven quality assurance methods for practical implementation. Also, be aware that applying our method to an unstable system can be counterproductive.

2. Because of the lower requirement for long-term memory in the two-tank system, the trade-off between recovery time and inference loss was not shown clearly enough in Fig.6. However, if tasks are dependent on long-term memory, the decline in the accuracy of recovery time estimation and the accuracy degradation caused by the penalty terms for the resilience become significant problems. As demonstrated in Theorem 5.4, the $\delta$ISS conditions form the foundation for the recovery time estimation; therefore, relaxing these conditions is necessary to address these problems. Approximating the limit of the sequence $A_s^{(l)}(k)$ and relaxing the $\delta$ISS condition could be a key approach. However, these methods require further investigation (see also Appendix B).

3. For continuous changes, such as environmental shifts and degradation over time, it is naturally expected that online learning will be addressed. While, as an idea for guaranteeing resilience without online learning, we believe that it can be extended to monitor the difference between the inferred and actual values, and to modify the weights based on the feedback of the difference into the LSTM (Schimperna et al., 2023). It may also be necessary to redefine recovery time flexibly.

4. Although our current research focused on the resilience of *LSTM modules* in control systems, comprehensive quality assurance for *control systems* remains a challenging issue. We consider that our framework can also be extended to feedback control, where the input depends on states. For example, in the classical closed-loop control where the input is given by $F(y_{target} - y_{out})$, the discussion using $U - WF$ instead of $U$ would extend our analysis. In MPC, Terzi et al. (2021) proved the convergence to the target values by incorporating $\delta$ISS LSTM as an observer and controller. Then, formulating the recovery time in the same way as this paper, it is expected to estimate the convergence time to target values when a disturbance occurs. By more precisely analyzing the calculations used in these studies to prove control system stability, it may be possible to determine the convergence time of the system state to the target value.

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

# A Proofs

## A.1 Proof of Proposition 4.1

We show the following lemma before proving Proposition 4.1.

**Lemma A.1** (Monotonic Decrease and Convergence). *Let $L \in \mathbb{N}$. $\{\overline{\sigma}_*^{(l)}(k)\}_{k=0}^{\infty}$, $\{\overline{\phi}_c^{(l)}(k)\}_{k=0}^{\infty}$, and $\{\eta^{(l)}(k)\}_{k=-1}^{\infty}$ are monotonically decreasing and converge for all $l \in [L]$, where $*$ denotes $f, i,$ or $o$.*

*Proof.* From the definition in Section 4.1, $0 \leq G_*^{(l)}(\eta^{(l)}(k-1)), G_c^{(l)}(\eta^{(l)}(k-1))$. Since $\sigma$ and $\phi$ are strictly monotonically increasing functions, we have $1/2 = \sigma(0) \leq \overline{\sigma}_*^{(l)}(k) < 1$ and $0 = \phi(0) \leq \overline{\phi}_c^{(l)}(k) < 1$. Therefore, $0 \leq \overline{\sigma}_i^{(l)}(k)\overline{\phi}_c^{(l)}(k)$ and $2 \leq 1/(1 - \overline{\sigma}_f^{(l)}(k))$ are obtained, then,

$$0 = \phi\left(0 \times 2\right)\frac{1}{2} \leq \phi\left(\frac{\overline{\sigma}_i^{(l)}(k)\overline{\phi}_c^{(l)}(k)}{1 - \overline{\sigma}_f^{(l)}(k)}\right)\overline{\sigma}_o^{(l)}(k) = \eta^{(l)}(k).$$

Hence, $\{\overline{\sigma}_*^{(l)}(k)\}_{k=0}^{\infty}$, $\{\overline{\phi}_c^{(l)}(k)\}_{k=0}^{\infty}$, and $\{\eta^{(l)}(k)\}_{k=-1}^{\infty}$ are bounded below sequences.

In the following, we show that these sequences are monotonically decreasing by induction, therefore converging.

Firstly, consider the case of $k = 0$. From the above inequalities, and the fact that $\sigma(x) \to \infty$ as $x \to \infty$, and that $0 < \overline{\sigma}_o^{(l)}(0) < 1$, we have

$$0 \leq \phi\left(\frac{\overline{\sigma}_i^{(l)}(0)\overline{\phi}_c^{(l)}(0)}{1 - \overline{\sigma}_f^{(l)}(0)}\right) < 1 = \eta^{(l)}(-1),$$

$$\eta^{(l)}(0) = \phi\left(\frac{\overline{\sigma}_i^{(l)}(0)\overline{\phi}_c^{(l)}(0)}{1 - \overline{\sigma}_f^{(l)}(0)}\right)\overline{\sigma}_o^{(l)}(0) \leq \overline{\sigma}_o^{(l)}(0) < \eta^{(l)}(-1).$$

Substituting $\eta^{(l)}(0) < \eta^{(l)}(-1)$ for $G_*(\eta)$ and $G_c(\eta)$, we obtain

$$G_*(\eta^{(l)}(0)) = \left\|\left(x_{\max}\left|W_*^{(l)}\right|\mathbf{1}_{n_x} + \eta^{(l)}(0)\left|U_*^{(l)}\right|\mathbf{1}_{n_c} + b_*^{(l)}\right)_+\right\|_{\infty}$$
$$\leq \left\|\left(x_{\max}\left|W_*^{(l)}\right|\mathbf{1}_{n_x} + \eta^{(l)}(-1)\left|U_*^{(l)}\right|\mathbf{1}_{n_c} + b_*^{(l)}\right)_+\right\|_{\infty} = G_*(\eta^{(l)}(-1)),$$

$$G_c(\eta^{(l)}(0)) = \left\|x_{\max}\left|W_c^{(l)}\right|\mathbf{1}_{n_x} + \eta^{(l)}(0)\left|U_c^{(l)}\right|\mathbf{1}_{n_c} + \left|b_c^{(l)}\right|\right\|_{\infty}$$
$$\leq \left\|x_{\max}\left|W_c^{(l)}\right|\mathbf{1}_{n_x} + \eta^{(l)}(-1)\left|U_c^{(l)}\right|\mathbf{1}_{n_c} + \left|b_c^{(l)}\right|\right\|_{\infty} = G_c(\eta^{(l)}(-1)).$$

Since $\sigma$ and $\phi$ are strictly monotonically increasing functions, the following hold:

$$\overline{\sigma}_*^{(l)}(1) = \sigma(G_*(\eta^{(l)}(0))) \leq \sigma(G_*(\eta^{(l)}(-1))) = \overline{\sigma}_k^{(l)}(0),$$
$$\overline{\phi}_c^{(l)}(1) = \phi(G_c(\eta^{(l)}(0))) \leq \phi(G_c(\eta^{(l)}(-1))) = \overline{\phi}_c^{(l)}(0).$$

Therefore, $\overline{\sigma}_*^{(l)}(1) \leq \overline{\sigma}_*^{(l)}(1)$ and $\overline{\phi}_c^{(l)}(1) \leq \overline{\phi}_c^{(l)}(0)$. Applying these inequality for $\overline{c}^{(l)}(1)$ and $\eta^{(l)}(1)$, we have

$$\overline{c}^{(l)}(1) = \frac{\overline{\sigma}_i^{(l)}(1)\overline{\phi}_c^{(l)}(1)}{1 - \overline{\sigma}_f^{(l)}(1)} \leq \frac{\overline{\sigma}_i^{(l)}(0)\overline{\phi}_c^{(l)}(0)}{1 - \overline{\sigma}_f^{(l)}(1)} \leq \frac{\overline{\sigma}_i^{(l)}(0)\overline{\phi}_c^{(l)}(0)}{1 - \overline{\sigma}_f^{(l)}(0)} = \overline{c}^{(l)}(0),$$

$$\eta^{(l)}(1) = \phi(\overline{c}^{(l)}(1))\overline{\sigma}_o^{(l)}(1) \leq \phi(\overline{c}^{(l)}(0))\overline{\sigma}_o^{(l)}(1) = \eta^{(l)}(0).$$

Thus, we obtain that $\bar{\sigma}_*^{(l)}(1) \leqslant \bar{\sigma}_*^{(l)}(0)$, $\bar{\phi}_c^{(l)}(1) \leqslant \bar{\phi}_c^{(l)}(0)$ and $\eta^{(l)}(1) \leqslant \eta^{(l)}(0) \leqslant \eta^{(l)}(-1) = 1$.

Secondly, consider the case of $k = k_0 \in \mathbb{Z}_{\geqslant 0}$ and assume that $\bar{\sigma}_*^{(l)}(k_0+1) \leqslant \bar{\sigma}_*^{(l)}(k_0)$, $\bar{\phi}_c^{(l)}(k_0+1) \leqslant \bar{\phi}_c^{(l)}(k_0)$, and $\eta^{(l)}(k_0+1) \leqslant \eta^{(l)}(k_0)$. From assumption, $G_*(\eta^{(l)}(k_0)) \leqslant G_*(\eta^{(l)}(k_0+1))$ and $G_c(\eta^{(l)}(k_0)) \leqslant G_c(\eta^{(l)}(k_0+1))$ hold in the same way as the case of $k = 0$. Therefore, $\bar{\sigma}_*^{(l)}(k_0+2) \leqslant \bar{\sigma}_*^{(l)}(k_0+1)$ and $\bar{\phi}_c^{(l)}(k_0+2) \leqslant \bar{\phi}_c^{(l)}(k_0+1)$ are obtained. By using these inequalities, we have

$$
\begin{aligned}
\bar{c}^{(l)}(k_0 + 2) &= \frac{\bar{\sigma}_i^{(l)}(k_0 + 2)\bar{\phi}_c^{(l)}(k_0 + 2)}{1 - \bar{\sigma}_f^{(l)}(k_0 + 2)} \\
&\leqslant \frac{\bar{\sigma}_i^{(l)}(k_0 + 1)\bar{\phi}_c^{(l)}(k_0 + 1)}{1 - \bar{\sigma}_f^{(l)}(k_0 + 2)} \leqslant \frac{\bar{\sigma}_i^{(l)}(k_0 + 1)\bar{\phi}_c^{(l)}(k_0 + 1)}{1 - \bar{\sigma}_f^{(l)}(k_0 + 1)} = \bar{c}^{(l)}(k_0 + 1).
\end{aligned}
$$

In addition, $\bar{\sigma}_o^{(l)}(k_0 + 2) \leqslant \bar{\sigma}_o^{(l)}(k_0 + 1)$ and $\phi$ is strictly monotonically increasing. Then, we obtain that $\eta(k_0 + 2) \leqslant \eta(k_0 + 1)$. $\qquad\square$

Here we restate and prove Proposition 4.1:

**Proposition A.2** (Invariant Set)**.** *Let $L \in \mathbb{N}$ and $k \in \mathbb{Z}_{\geqslant 0}$. $\mathcal{S}^{(l)}(k) \coloneqq \mathcal{C}^{(l)}(k) \times \mathcal{H}^{(l)}(k)$ is an invariant set, that is, for all $t \in \mathbb{Z}_{\geqslant 0}$, $l \in [L]$, and $x \in [-x_{\max}, x_{\max}]^{n_x}$, if $s^{(l)}(0) = (c^{(l)}(0)^\top, h^{(l)}(0)^\top)^\top \in \mathcal{S}^{(l)}(k)$, then $s^{(l)}(t) = (c^{(l)}(t)^\top, h^{(l)}(t)^\top)^\top \in \mathcal{S}^{(l)}(k)$. Furthermore, the sequence $\{\mathcal{S}^{(l)}(k)\}_{k=0}^{\infty}$ is a decreasing sequence with respect to $k$ in the sense of set inclusion.*

*Proof.* From Lemma A.1 and the definition of $\mathcal{C}^{(l)}(k)$ and $\mathcal{H}^{(l)}(k)$ in Section 4.1, it is obvious that $\{\mathcal{S}^{(l)}(k)\}_k$ is a decreasing sequence.

Below, we show $\{\mathcal{S}^{(l)}(k)\}_k$ is an invariant set. Assume $s^{(l)}(0) = (c^{(l)}(0)^\top, h^{(l)}(0)^\top)^\top \in \mathcal{S}^{(l)}(k)$, then we need only to check that

$$
s^{(l)}(t) = (c^{(l)}(t)^\top, h^{(l)}(t)^\top)^\top \in \mathcal{S}^{(l)}(k) \implies s^{(l)}(t + 1) = (c^{(l)}(t + 1)^\top, c^{(l)}(t + 1)^\top)^\top \in \mathcal{S}^{(l)}(k)
$$

for all $t \in \mathbb{Z}_{\geqslant 0}$ and $x \in [-x_{\max}, x_{\max}]^{n_x}$.

First, we assume $c^{(l)}(t) \in \mathcal{C}^{(l)}(k)$. Since $h^{(l)}(t) \in \mathcal{H}^{(l)}(k)$,

$$
\|h^{(l)}(t)\|_\infty \leqslant \phi(\bar{c}^{(l)}(k))\bar{\sigma}_o^{(l)}(k) = \eta^{(l)}(k).
$$

Therefore, for all $j \in \{1, \ldots, n_c\}$, it hold that

$$
\begin{aligned}
\sigma(W_*^{(l)} x^{(l)}(t) + U_*^{(l)} h^{(l)}(t) + b_*^{(l)})_{(j)} &\leqslant \sigma\left(\left\|(x_{\max}\left|W_*^{(l)}\right|\mathbf{1}_{n_x} + \left|U_*^{(l)}\right|h^{(l)}(t) + b_*^{(l)})_+\right\|_\infty\right) \\
&= \sigma\left(\left\|(x_{\max}\left|W_*^{(l)}\right|\mathbf{1}_{n_x} + \eta^{(l)}(k)\left|U_*^{(l)}\right|\mathbf{1}_{n_c} + b_*^{(l)})_+\right\|_\infty\right) \\
&\leqslant \sigma\left(\left\|(x_{\max}\left|W_*^{(l)}\right|\mathbf{1}_{n_x} + \eta^{(l)}(k-1)\left|U_*^{(l)}\right|\mathbf{1}_{n_c} + b_*^{(l)})_+\right\|_\infty\right) \\
&= \sigma(G_*^{(l)}(\eta^{(l)}(k-1))) \\
&= \bar{\sigma}_*^{(l)}(k),
\end{aligned}
$$

where $*$ denotes $f, i$ or $o$. We have $\phi(W_c^{(l)}x^{(l)}(t) + U_c^{(l)}h^{(l)}(t) + b_c^{(l)})_{(j)} \leqslant \overline{\phi}_c^{(l)}(k)$ in the same way. Thus, each component of $c^{(l)}(t+1)$ is evaluated as follows:

$$
\begin{aligned}
c^{(l)}(t+1)_{(j)} &= \sigma(W_f^{(l)}x^{(l)}(t) + U_f^{(l)}h^{(l)}(t) + b_f^{(l)})_{(j)} \times c^{(l)}(t)_{(j)} \\
&\quad + \sigma(W_i^{(l)}x^{(l)}(t) + U_i^{(l)}h^{(l)}(t) + b_i^{(l)})_{(j)} \times \phi(W_c^{(l)}x^{(l)}(t) + U_c^{(l)}h^{(l)}(t) + b_c^{(l)})_{(j)} \\
&\leqslant \overline{\sigma}_f^{(l)}(k)\|c^{(l)}(t)\|_\infty + \overline{\sigma}_i^{(l)}(k)\overline{\phi}_c^{(l)}(k) \\
&= \overline{\sigma}_f^{(l)}(k)\frac{\overline{\sigma}_i^{(l)}(k)\overline{\phi}_c^{(l)}(k)}{1 - \overline{\sigma}_f^{(l)}(k)} + \overline{\sigma}_i^{(l)}(k)\overline{\phi}_c^{(l)}(k) \\
&= \overline{c}^{(l)}(k).
\end{aligned}
$$

This inequality holds for all $j \in \{1, \ldots, n_c\}$, therefore $c^{(l)}(t+1) \in \mathcal{C}^{(l)}(k)$.

Next, we assume $h^{(l)}(t) \in \mathcal{H}^{(l)}(k)$. Since hyperbolic tangent $\phi$ is strictly monotonically increasing,

$$
\begin{aligned}
h^{(l)}(t+1)_{(j)} &= \sigma(W_o^{(l)}x^{(l)}(t) + U_o^{(l)}h^{(l)}(t) + b_o^{(l)})_{(j)} \times \phi(c^{(l)}(t+1))_{(j)} \\
&\leqslant \overline{\sigma}_o^{(l)}(k)\phi(\overline{c}^{(l)}(k)).
\end{aligned}
$$

This inequality also holds for all $j \in \{1, \ldots, n_c\}$, therefore $h^{(l)}(t+1) \in \mathcal{H}^{(l)}(k)$.

Hence, we conclude that $s^{(l)}(t+1) \in \mathcal{S}^{(l)}(k)$ when $s^{(l)}(t) \in \mathcal{S}^{(l)}(k)$. $\qquad\square$

## A.2 Proof of Theorem 4.2

In this section, we prove the input-to-state stability and the incremental input-to-state stability of LSTM.

First, we introduce the definition of input-to-state stability.

**Definition A.3** (ISS, Sontag 1990; Jiang & Wang 2001)**.** System (1) is called input-to-state stable (ISS) if there exist continuous functions $\beta \in \mathcal{KL}$ and $\gamma \in \mathcal{K}_\infty$ such that for any initial state $s(0) \in \mathbb{R}^{n_s}$ and for any input sequence $x(0:t) \in [-x_{\max}, x_{\max}]^{n_x \times (t+1)}$, $s(t)$ satisfies

$$
\|s(t)\| \leqslant \beta(\|s(0)\|, t) + \gamma(\|x(0:t)\|_{2,\infty})
$$

for all $t \in \mathbb{Z}_{\geqslant 0}$.

Considering the invariant set in Proposition 4.1, the sufficient condition for ISS of LSTM is more relaxed (in the case of $k \geqslant 1$) than (Terzi et al., 2021, Theorem 1, which is the case of $k = 0$) as follows.

**Theorem A.4** (ISS of LSTM)**.** *Let* $L \in \mathbb{N}$ *and* $k \in \mathbb{Z}_{\geqslant 0}$. *We assume that* $s^{(l)}(0) \in \mathcal{S}^{(l)}(k)$ *for all* $l \in [L]$. *The LSTM is ISS if* $\rho(A^{(l)}(k)) < 1$ *for all* $l \in [L]$, *where*

$$
A_s^{(l)}(k) = \begin{bmatrix} \overline{\sigma}_f(k) & \overline{\sigma}_i(k)\|U_c\| \\ \overline{\sigma}_o(k)\overline{\sigma}_f(k) & \overline{\sigma}_o(k)\overline{\sigma}_i(k)\|U_c\| \end{bmatrix}.
$$

*Furthermore,* $\rho(A_s^{(l)}(k))$ *is monotonically decreasing with respect to* $k \in \mathbb{Z}_{\geqslant 0}$.

*Proof.* From (2a) and the fact Lipschitz constant of $\phi$ is 1, we have

$$
\begin{aligned}
\left\| c^{(l)}(t+1) \right\| &\leqslant \left\| \sigma \left( W_f^{(l)} x^{(l)}(t) + U_f^{(l)} h^{(l)}(t) + b_f^{(l)} \right) \right\|_\infty \left\| c^{(l)}(t) \right\| \\
&\quad + \left\| \sigma \left( W_i^{(l)} x^{(l)}(t) + U_i^{(l)} h^{(l)}(t) + b_i^{(l)} \right) \right\|_\infty \left\| \phi \left( W_c^{(l)} x^{(l)}(t) + U_c^{(l)} h^{(l)}(t) + b_c^{(l)} \right) \right\| \\
&\leqslant \sigma \left( \left\| W_f^{(l)} x^{(l)}(t) + U_f^{(l)} h^{(l)}(t) + b_f^{(l)} \right\|_\infty \right) \left\| c^{(l)}(t) \right\| \\
&\quad + \sigma \left( \left\| W_i^{(l)} x^{(l)}(t) + U_i^{(l)} h^{(l)}(t) + b_i^{(l)} \right\|_\infty \right) \left\| W_c^{(l)} x^{(l)}(t) + U_c^{(l)} h^{(l)}(t) + b_c^{(l)} \right\| \\
&\leqslant \sigma \left( G_f^{(l)}(\eta^{(l)}(k-1)) \right) \left\| c^{(l)}(t) \right\| \\
&\quad + \sigma \left( G_i^{(l)}(\eta^{(l)}(k-1)) \right) \left( \left\| W_c^{(l)} \right\| \left\| x^{(l)}(t) \right\| + \left\| U_c^{(l)} \right\| \left\| h^{(l)}(t) \right\| + \left\| b_c^{(l)} \right\| \right) \\
&= \overline{\sigma}_f^{(l)}(k) \left\| c^{(l)}(t) \right\| + \overline{\sigma}_i^{(l)}(k) \left( \left\| W_c^{(l)} \right\| \left\| x^{(l)}(t) \right\| + \left\| U_c^{(l)} \right\| \left\| h^{(l)}(t) \right\| + \left\| b_c^{(l)} \right\| \right)
\end{aligned}
$$

for any $k \in \mathbb{Z}_{\geqslant 0}$ and $l \in [L]$. Similarly, (2b) implies that

$$
\begin{aligned}
\left\| h^{(l)}(t+1) \right\| &\leqslant \left\| \sigma \left( W_o^{(l)} x^{(l)}(t) + U_o^{(l)} h^{(l)}(t) + b_o^{(l)} \right) \right\|_\infty \left\| \phi \left( c^{(l)}(t+1) \right) \right\| \\
&\leqslant \sigma \left( \left\| W_o^{(l)} x^{(l)}(t) + U_o^{(l)} h^{(l)}(t) + b_o^{(l)} \right\|_\infty \right) \left\| c^{(l)}(t+1) \right\| \\
&\leqslant \overline{\sigma}_o^{(l)}(k) \left( \overline{\sigma}_f^{(l)}(k) \left\| c^{(l)}(t) \right\| + \overline{\sigma}_i^{(l)}(k) \left( \left\| W_c^{(l)} \right\| \left\| x^{(l)}(t) \right\| + \left\| U_c^{(l)} \right\| \left\| h^{(l)}(t) \right\| + \left\| b_c^{(l)} \right\| \right) \right).
\end{aligned}
$$

Then,

$$
\begin{aligned}
\begin{pmatrix} \left\| c^{(l)}(t+1) \right\| \\ \left\| h^{(l)}(t+1) \right\| \end{pmatrix} &\leqslant \begin{bmatrix} \overline{\sigma}_f^{(l)}(k) & \overline{\sigma}_i^{(l)}(k) \left\| U_c^{(l)} \right\| \\ \overline{\sigma}_o^{(l)}(k)\overline{\sigma}_f^{(l)}(k) & \overline{\sigma}_o^{(l)}(k)\overline{\sigma}_i^{(l)}(k) \left\| U_c^{(l)} \right\| \end{bmatrix} \begin{pmatrix} \left\| c^{(l)}(t) \right\| \\ \left\| h^{(l)}(t) \right\| \end{pmatrix} \\
&\quad + \begin{pmatrix} \overline{\sigma}_i^{(l)}(k) \left\| W_c^{(l)} \right\| \\ \overline{\sigma}_o^{(l)}(k)\overline{\sigma}_i^{(l)}(k) \left\| W_c^{(l)} \right\| \end{pmatrix} \left\| x^{(l)} \right\| + \begin{pmatrix} \overline{\sigma}_i^{(l)}(k) \\ \overline{\sigma}_o^{(l)}(k)\overline{\sigma}_i^{(l)}(k) \end{pmatrix} \left\| b_c^{(l)} \right\| \\
&=: A_s^{(l)}(k) \begin{pmatrix} \left\| c^{(l)}(t) \right\| \\ \left\| h^{(l)}(t) \right\| \end{pmatrix} + a_x^{(l)}(k) \left\| x^{(l)} \right\| + a_b^{(l)}(k) \left\| b_c^{(l)} \right\|.
\end{aligned}
$$

For each layer $l \in [L]$ of LSTM, we estimate for any $k \in \mathbb{Z}_{\geqslant 0}$ that

$$
\begin{aligned}
\begin{pmatrix} \left\| c^{(l)}(t) \right\| \\ \left\| h^{(l)}(t) \right\| \end{pmatrix} &\leqslant A_s^{(l)}(k) \begin{pmatrix} \left\| c^{(l)}(t-1) \right\| \\ \left\| h^{(l)}(t-1) \right\| \end{pmatrix} + a_x^{(l)}(k) \left\| x^{(l)}(t-1) \right\| + a_b^{(l)}(k) \left\| b_c^{(l)} \right\| \\
&\leqslant A_s^{(l)}(k)^2 \begin{pmatrix} \left\| c^{(l)}(t-2) \right\| \\ \left\| h^{(l)}(t-2) \right\| \end{pmatrix} + A_s^{(l)}(k)a_x^{(l)}(k) \left\| x^{(l)}(t-2) \right\| + A_s^{(l)}(k)a_b^{(l)}(k) \left\| b_c^{(l)} \right\| \\
&\quad + a_x^{(l)}(k) \left\| x^{(l)}(t-1) \right\| + a_b^{(l)}(k) \left\| b_c^{(l)} \right\| \\
&\vdots \\
&\leqslant A_s^{(l)}(k)^t \begin{pmatrix} \left\| c^{(l)}(0) \right\| \\ \left\| h^{(l)}(0) \right\| \end{pmatrix} + \sum_{\tau=0}^{t-1} A_s^{(l)}(k)^{t-\tau-1} a_x^{(l)}(k) \left\| x^{(l)}(\tau) \right\| + \left\| b_c^{(l)} \right\| \sum_{\tau=0}^{t-1} A_s^{(l)}(k)^\tau a_b^{(l)}(k), \quad (9)
\end{aligned}
$$

where $a_b^{(l)} := 0$ for $l = 1, \ldots, L-1$.

For single-layer LSTM (the case of $L = 1$), we obtain that for constant $\mu > 1$,

$$
\begin{aligned}
\|s(t)\| = \left\| \begin{pmatrix} c(t) \\ h(t) \end{pmatrix} \right\| &= \sqrt{\|c(t)\|^2 + \|h(t)\|^2} = \left\| \begin{pmatrix} \|c(t)\| \\ \|h(t)\| \end{pmatrix} \right\| \\
&\leqslant \left\| A_s(k)^t \right\| \|s(0)\| + \left\| \sum_{j=0}^{t-1} A_s(k)^{t-j-1} a_x(k) \|x(j)\| \right\| + \|b_c\| \left\| \sum_{j=0}^{t-1} A_s(k)^j a_b(k) \right\| \qquad (10) \\
&\leqslant \left\| A_s(k)^t \right\| \|s(0)\| + \left\| \sum_{j=0}^{t-1} A_s(k)^j a_x(k) \right\| \max_{j \in [t]} \|x(j-1)\| + \left\| \sum_{j=0}^{t-1} A_s(k)^j a_b(k) \right\| \|b_c\| \\
&\leqslant \mu \lambda_1(A_s(k))^t \|s(0)\| \\
&\quad + \left\| (I - A_s(k)^t)(I - A_s(k))^{-1} a_x(k) \right\| \|x\|_{2,\infty} + \left\| (I - A_s(k))^{-1} a_b(k) \right\| \|b_c\|. \qquad (11)
\end{aligned}
$$

Here, we note that $\lambda_1(A_s(k)) \in (0, 1)$ since $\rho(A_s(k)) < 1$. Therefore, the first term of (11) belongs to class $\mathcal{KL}(\|s(0)\|, t)$, the second term belongs to class $\mathcal{K}_\infty(\|x\|_{2,\infty})$, and the third term belongs to class $\mathcal{K}_\infty(\|b_c\|)$. This concludes that single-layer LSTM is ISS if $\rho(A_s(k)) < 1$.

For multi-layer LSTM (the case of $L \geqslant 2$), it follows from (10) that

$$
\begin{aligned}
\left\| s^{(L)}(t) \right\| &\leqslant \left\| A_s^{(L)}(k)^t \right\| \left\| s^{(L)}(0) \right\| \\
&\quad + \left\| \sum_{\tau=0}^{t-1} A_s^{(L)}(k)^{t-\tau-1} a_x^{(L)}(k) \left\| x^{(L)}(\tau) \right\| \right\| + \left\| b_c^{(L)} \right\| \left\| \sum_{\tau=0}^{t-1} A_s^{(L)}(k)^\tau a_b^{(L)}(k) \right\|. \qquad (12)
\end{aligned}
$$

Also, the estimate (9) imply that

$$
\left\| h^{(l)}(t) \right\| \leqslant \left\llcorner A_s^{(l)}(k)^t \right\lrcorner \left( \begin{matrix} \left\| c^{(l)}(0) \right\| \\ \left\| h^{(l)}(0) \right\| \end{matrix} \right) + \sum_{\tau=0}^{t-1} \left\llcorner A_s^{(l)}(k)^{t-\tau-1} \right\lrcorner a_x^{(l)}(k) \left\| x^{(l)}(\tau) \right\| \qquad (13)
$$

for $l = 1, \ldots, L-1$, where $\llcorner A \lrcorner$ denotes the lower horizontal vector $(A_{(2,1)}, A_{(2,2)})$ for a matrix $A = \begin{bmatrix} A_{(1,1)} & A_{(1,2)} \\ A_{(2,1)} & A_{(2,2)} \end{bmatrix}$. Applying the relation (2d) and inequality (13) to the second term of (12) repeatedly,

we estimate that

$$
\left\| \sum_{\tau=0}^{t-1} A_s^{(L)}(k)^{t-\tau-1} a_x^{(L)}(k) \left\| x^{(L)}(\tau) \right\| \right\| = \left\| \sum_{\tau_L=1}^{t} A_s^{(L)}(k)^{t-\tau_L} a_x^{(L)}(k) \left\| h^{(L-1)}(\tau_L) \right\| \right\|
$$

$$
\leqslant \left\| \sum_{\tau_L=1}^{t} A_s^{(L)}(k)^{t-\tau_L} a_x^{(L)}(k) \llcorner A_s^{(L-1)}(k)^{\tau_L} \lrcorner \right\| \left\| s^{(L-1)}(0) \right\|
$$

$$
+ \left\| \sum_{\tau_L=1}^{t} A_s^{(L)}(k)^{t-\tau_L} a_x^{(L)}(k) \sum_{\tau_{L-1}=1}^{\tau_L} \llcorner A_s^{(L-1)}(k)^{\tau_L-\tau_{L-1}} a_x^{(L-1)}(k) \left\| h^{(L-2)}(\tau_{L-1}) \right\| \right\|
$$

$$
\leqslant \left\| \sum_{\tau_L=1}^{t} A_s^{(L)}(k)^{t-\tau_L} a_x^{(L)}(k) \llcorner A_s^{(L-1)}(k)^{\tau_L} \lrcorner \right\| \left\| s^{(L-1)}(0) \right\|
$$

$$
+ \left\| \sum_{\tau_L=1}^{t} A_s^{(L)}(k)^{t-\tau_L} a_x^{(L)}(k) \sum_{\tau_{L-1}=1}^{\tau_L} \llcorner A_s^{(L-1)}(k)^{\tau_L-\tau_{L-1}} a_x^{(L-1)}(k) \llcorner A_s^{(L-2)}(k)^{\tau_{L-1}} \lrcorner \right\| \left\| s^{(L-2)}(0) \right\|
$$

$$
\vdots
$$

$$
+ \left\| \sum_{\tau_L=1}^{t} A_s^{(L)}(k)^{t-\tau_L} a_x^{(L)}(k) \sum_{\tau_{L-1}=1}^{\tau_L} \llcorner A_s^{(L-1)}(k)^{\tau_L-\tau_{L-1}} a_x^{(L-1)}(k) \cdots \right.
$$

$$
\left. \cdots \sum_{\tau_2=1}^{\tau_3} \llcorner A_s^{(2)}(k)^{\tau_3-\tau_2} a_x^{(2)}(k) \llcorner A_s^{(1)}(k)^{\tau_2} \lrcorner \right\| \left\| s^{(1)}(0) \right\|
$$

$$
+ \left\| \sum_{\tau_L=1}^{t} A_s^{(L)}(k)^{t-\tau_L} a_x^{(L)}(k) \sum_{\tau_{L-1}=1}^{\tau_L} \llcorner A_s^{(L-1)}(k)^{\tau_L-\tau_{L-1}} a_x^{(L-1)}(k) \cdots \right.
$$

$$
\left. \cdots \sum_{\tau_1=1}^{\tau_2} \llcorner A_s^{(1)}(k)^{\tau_2-\tau_1} a_x^{(1)}(k) \left\| x(\tau_1 - 1) \right\| \right\|
$$

$$
= \left\| \sum_{\tau_L=1}^{t} A_s^{(L)}(k)^{t-\tau_L} a_x^{(L)}(k) \llcorner A_s^{(L-1)}(k)^{\tau_L} \lrcorner \right\| \left\| s^{(L-1)}(0) \right\|
$$

$$
+ \sum_{m=1}^{L-2} \left\| \sum_{\tau_L=1}^{t} A_s^{(L)}(k)^{t-\tau_L} a_x^{(L)}(k) \sum_{\tau_{L-1}=1}^{\tau_L} \llcorner A_s^{(L-1)}(k)^{\tau_L-\tau_{L-1}} a_x^{(L-1)}(k) \sum_{\tau_{L-2}=1}^{\tau_{L-1}} \cdots \right.
$$

$$
\left. \cdots \sum_{\tau_{m+1}=1}^{\tau_{m+2}} \llcorner A_s^{(m+1)}(k)^{\tau_{m+2}-\tau_{m+1}} a_x^{(m+1)}(k) \llcorner A_s^{(m)}(k)^{\tau_{m+1}} \lrcorner \right\| \left\| s^{(m)}(0) \right\|
$$

$$
+ \left\| \sum_{\tau_L=1}^{t} A_s^{(L)}(k)^{t-\tau_L} a_x^{(L)}(k) \sum_{\tau_{L-1}=1}^{\tau_L} \llcorner A_s^{(L-1)}(k)^{\tau_L-\tau_{L-1}} a_x^{(L-1)}(k) \sum_{\tau_{L-2}=1}^{\tau_{L-1}} \cdots \right.
$$

$$
\left. \cdots \sum_{\tau_1=1}^{\tau_2} \llcorner A_s^{(1)}(k)^{\tau_2-\tau_1} a_x^{(1)}(k) \left\| x(\tau_1 - 1) \right\| \right\|
$$

$$
\leqslant \sum_{\tau=1}^{t} \left\| A_s^{(L)}(k)^{(t-\tau)} \right\| \left\| a_x^{(L)} \right\| \left\| \llcorner A_s^{(L-1)}(k)^{\tau} \lrcorner \right\| \left\| s^{(L-1)}(0) \right\| + \sum_{m=1}^{L-2} \left( \sum_{\tau_L=1}^{t} \sum_{\tau_{L-1}=1}^{\tau_L} \cdots \right.
$$

$$
\left. \cdots \sum_{\tau_{m+1}=1}^{\tau_{m+2}} \left\| A_s^{(L)}(k)^{t-\tau_L} \right\| \prod_{l=m+1}^{L-1} \left\| \llcorner A_s^{(l)}(k)^{\tau_{l+1}-\tau_l} \lrcorner \right\| \left\| A_s^{(m)}(k)^{\tau_{m+1}} \right\| \prod_{j=m+1}^{L} \left\| a_x^{(j)}(k) \right\| \right) \left\| s^{(m)}(0) \right\|
$$

$$
+ \sum_{\tau_L=1}^{t} \sum_{\tau_{L-1}=1}^{\tau_L} \cdots \sum_{\tau_1=1}^{\tau_2} \left\| A_s^{(L)}(k)^{t-\tau_L} \right\| \prod_{l=1}^{L-1} \left\| \llcorner A_s^{(l)}(k)^{\tau_{l+1}-\tau_l} \lrcorner \right\| \prod_{j=1}^{L} \left\| a_x^{(j)}(k) \right\| \left\| x(\tau_1 - 1) \right\|,
$$

where the last inequality follows from the inequality $\|Aa_1\| \leqslant \|A\| \|a_1\|$ and equality $\|a_1 a_2^\top\| = \max_{|\xi|=1} \|a_1 a_2 \xi\| = \|a_1\| \max_{|\xi|=1} |a_2 \xi| = \|a_1\| \|a_2\|$ for matrix $A \in \mathbb{R}^{2\times2}$ and vectors $a_1, a_2 \in \mathbb{R}^2$. Then, we estimate (12) again that

$$\left\| s^{(L)}(t) \right\|$$

$$\leqslant \left\| A_s^{(L)}(k)^t \right\| \left\| x^{(L)}(0) \right\| + \sum_{m=1}^{L-1} \left( \sum_{\sum_{l=m}^{L} \tau_l = t, \tau_l \geqslant 0} \prod_{l=m}^{L} \left\| A_s^{(l)}(k)^{\tau_l} \right\| \prod_{l=m+1}^{L} \left\| a_x^{(l)}(k) \right\| \right) \left\| s^{(m)}(0) \right\|$$

$$+ \sum_{\tau=0}^{t-1} \sum_{\sum_{l=1}^{L} \tau_l = t-\tau-1, \tau_l \geqslant 0} \left( \prod_{l=1}^{L} \left\| A_s^{(l)}(k)^{\tau_l} \right\| \left\| a_x^{(l)} \right\| \right) \|x(\tau)\| + \sum_{\tau=0}^{t-1} \left\| A_s^{(L)}(k)^\tau \right\| \left\| a_b^{(L)} \right\| \left\| b_c^{(L)} \right\|$$

$$\leqslant \mu^{(L)}(t) \rho(A_s^{(L)}(k))^t \left\| s^{(L)}(0) \right\| + \sum_{m=1}^{L-1} \left( \sum_{\sum_{l=m}^{L} \tau_l = t} \prod_{l=m}^{L} \mu^{(l)} \rho(A_s^{(l)}(k))^{\tau_l} \right) \left( \prod_{l=m+1}^{L} \left\| a_x^{(l)}(k) \right\| \right) \left\| s^{(m)}(0) \right\|$$

$$+ \sum_{\tau=0}^{t-1} \left( \sum_{\sum_{l=1}^{L} \tau_l = t-\tau-1} \prod_{l=1}^{L} \mu^{(l)} \rho(A_s^{(l)}(k))^{\tau_l} \right) \left( \prod_{l=1}^{L} \left\| a_x^{(l)}(k) \right\| \right) \|x(\tau)\| + \sum_{\tau=0}^{t-1} \mu^{(L)} \rho(A_s^{(L)}(k))^\tau \left\| a_b^{(L)} \right\| \left\| b_c^{(L)} \right\|$$

$$\leqslant \mu^{(L)}(t) \rho(A_s^{(L)}(k))^t \left\| s^{(L)}(0) \right\|$$

$$+ \sum_{m=1}^{L-1} \mu^{(m:L)}(t) \frac{(t+L-m)!}{t!(L-m)!} \max_{l=m,\dots,L} \rho(A_s^{(l)}(k))^t \left( \prod_{l=m+1}^{L} \left\| a_x^{(l)}(k) \right\| \right) \left\| s^{(m)}(0) \right\|$$

$$+ \sum_{\tau=0}^{t-1} \mu^{(1:L)}(t) \frac{(t+L-\tau-2)!}{(t-\tau-1)!(L-1)!} \max_{l=1,\dots,L} \rho(A_s^{(l)}(k))^{t-\tau-1} \left( \prod_{l=1}^{L} \left\| a_x^{(l)}(k) \right\| \right) \|x(\tau)\|$$

$$+ \sum_{\tau=0}^{t-1} \mu^{(L)}(t) \rho(A_s^{(L)}(k))^\tau \left\| a_b^{(L)} \right\| \left\| b_c^{(L)} \right\|$$

$$\leqslant M_L(k) \rho_L(k)^t \left\| s^{(1:L)}(0) \right\|_{2,\infty} + M_L'(k) \sum_{\tau=0}^{t-1} \rho_L'(k)^{t-\tau-1} \|x\|_{2,\infty} + M_L''(k) \sum_{\tau=0}^{t-1} \rho(A_s^{(L)}(k))^\tau \left\| b_c^{(L)} \right\|,$$

where $M_L(k), M_L'(k)$, and $M_L''(k)$ are constants independent of $t$, $0 < \rho_L(k), \rho_L'(k) < 1$, and $s^{(1:L)}(t) := [s^{(1)}(t), \dots, s^{(L)}(t)]$. This is because $\rho(A_s^l(k)) < \rho(A_s^L(k))$ and the products of $\mu$ and $\rho(A_s^L(k))$ are converge. Therefore, if $\rho(A_s^{(l)}(k)) \in (0,1)$ for any $l \in [L]$, the multi-layer LSTM is ISS.

We note that $\mu^{(l)}$ can be expressed by direct calculation as

$$\left\| A_s^{(l)}(k)^t \right\|$$

$$= \left\| U \begin{bmatrix} \lambda_1^{(l)} & \nu^{(l)} \\ 0 & \lambda_2^{(l)} \end{bmatrix} U^* \right\| = \left\| \begin{bmatrix} (\lambda_1^{(l)})^t & \nu^{(l)} \sum_{\tau=1}^{t} (\lambda_1^{(l)})^{t-\tau}(\lambda_2^{(l)})^{\tau-1} \\ 0 & (\lambda_2^{(l)})^t \end{bmatrix} \right\|$$

$$\leqslant \frac{1}{\sqrt{2}} \left\{ (1 + (r^{(l)})^{2t})(\lambda_1^{(l)})^2 + (\nu^{(l)})^2 \sum_{\tau=1}^{t} (r^{(l)})^{\tau-1} \sum_{v=1}^{t} (r^{(l)})^{v-1} \right\} (\lambda_1^{(l)})^{2(t-1)}$$

$$+ \left[ \left\{ (1 + (r^{(l)})^{2t})(\lambda_1^{(l)})^2 + (\nu^{(l)})^2 \sum_{\tau=1}^{t} (r^{(l)})^{\tau-1} \sum_{v=1}^{t} (r^{(l)})^{v-1} \right\}^2 - 4(r^{(l)})^{2t}(\lambda_1^{(l)})^4 \right]^{\frac{1}{2}} (\lambda_1^{(l)})^{2(t-1)}$$

$$\leqslant \rho(A_s^{(l)}(k))^t \sqrt{(1 + (r^{(l)})^{2t}) + \frac{|\nu^{(l)}|^2}{|\lambda_1^{(l)}|^2} \left( \frac{1 - (r^{(l)})^t}{1 - r^{(l)}} \right)^2}.$$

Finally, we show $\rho(A_s^{(l)}(k))$ is monotonically decreasing. Suppose that $\rho(A_s^{(l)}(k)) < \rho(A_s^{(l)}(k+1))$. Since $A_s^{(l)}(k)$ and $A_s^{(l)}(k+1)$ are non-negative matrices, $\rho(A_s^{(l)}(k))$ and $\rho(A_s^{(l)}(k+1))$ are Perron-Frobenius eigenvalues of $A_s^{(l)}(k)$ and $A_s^{(l)}(k+1)$, respectively. Also, there exists a non-negative eigenvector $v$ of $A_s^{(l)}(k+1)$ for eigenvalue $\rho(A_s^{(l)}(k+1))$. From Lemma A.1, $A_s^{(l)}(k) - A_s^{(l)}(k+1)$ is non-negative matrix. Then, the vector

$$-(A_s^{(l)}(k) - A_s^{(l)}(k+1))v = \rho(A_s^{(l)}(k+1))v - A_s^{(l)}(k)v = (\rho(A_s^{(l)}(k+1))I - A_s^{(l)}(k))v \tag{14}$$

is non-positive. Since all eigenvalues of $A_s^{(l)}(k)$ less than $\rho(A_s^{(l)}(k+1))$, then $\rho(A_s^{(l)}(k+1))I - A_s^{(l)}(k)$ is a regular matrix, and which inverse matrix can be expanded as

$$(\rho(A_s^{(l)}(k+1))I - A_s^{(l)}(k))^{-1} = \frac{1}{\rho(A_s^{(l)}(k+1))} \sum_{q=0}^{\infty} \left( \frac{A_s^{(l)}(k)}{\rho(A_s^{(l)}(k+1))} \right)^q.$$

Therefore, $(\rho(A_s^{(l)}(k+1))I - A_s^{(l)}(k))^{-1}$ is positive matrix. Hence, $v$ is non-positive from (14). Since $v$ is also a non-negative vector, $v = 0$. This contradicts that $v$ is an eigenvector. Thus, we conclude that $\rho(A_s^{(l)}(k)) \geqslant \rho(A_s^{(l)}(k+1))$. $\qquad\square$

The next is the same as Theorem 4.2.

**Theorem A.5** ($\delta$ISS of LSTM)**.** *Let $L \in \mathbb{N}$ and $k \in \mathbb{Z}_{\geqslant 0}$. We assume that $s^{(l)}(0) \in \mathcal{S}^{(l)}(k)$ for all $l \in [L]$. The LSTM is $\delta$ISS if $\rho(A_s^{(l)}(k)) < 1$ for all $l \in [L]$, where*

$$A_s^{(l)}(k) = \begin{bmatrix} \overline{\sigma}_f^{(l)}(k) & \alpha_s^{(l)}(k) \\ \overline{\sigma}_o^{(l)}(k)\overline{\sigma}_f^{(l)}(k) & \alpha_s^{(l)}(k)\overline{\sigma}_o^{(l)}(k) + \frac{1}{4}\phi(\overline{c}^{(l)}(k))\|U_o^{(l)}\| \end{bmatrix},$$

$$\alpha_s^{(l)}(k) = \frac{1}{4}\|U_f^{(l)}\|\overline{c}^{(l)}(k) + \overline{\sigma}_i^{(l)}(k)\|U_c^{(l)}\| + \frac{1}{4}\|U_i^{(l)}\|\overline{\phi}_c^{(l)}(k).$$

*Furthermore, $\rho(A_s^{(l)}(k))$ is monotonically decreasing with respect to $k \in \mathbb{Z}_{\geqslant 0}$.*

*Proof.* From (2a) and (2b), we have

$$\begin{aligned}
&c_1^{(l)}(t+1) - c_2^{(l)}(t+1) \\
&= \sigma\left(W_f^{(l)}x_1^{(l)}(t) + U_f^{(l)}h_1^{(l)}(t) + b_f^{(l)}\right) \odot \left(c_1^{(l)}(t) - c_2^{(l)}(t)\right) \\
&\quad + \left\{\sigma\left(W_f^{(l)}x_1^{(l)}(t) + U_f^{(l)}h_1^{(l)}(t) + b_f^{(l)}\right) - \sigma\left(W_f^{(l)}x_2^{(l)}(t) + U_f^{(l)}h_2^{(l)}(t) + b_f^{(l)}\right)\right\} \odot c_2^{(l)}(t) \\
&\quad + \sigma\left(W_i^{(l)}x_1^{(l)}(t) + U_i^{(l)}h_1^{(l)}(t) + b_i^{(l)}\right) \\
&\qquad \odot \left\{\phi\left(W_c^{(l)}x_1^{(l)}(t) + U_c^{(l)}h_1^{(l)}(t) + b_c^{(l)}\right) - \phi\left(W_c^{(l)}x_2^{(l)}(t) + U_c^{(l)}h_2^{(l)}(t) + b_c^{(l)}\right)\right\} \\
&\quad + \left\{\sigma\left(W_i^{(l)}x_1^{(l)}(t) + U_i^{(l)}h_1^{(l)}(t) + b_i^{(l)}\right) - \sigma\left(W_i^{(l)}x_2^{(l)}(t) + U_i^{(l)}h_2^{(l)}(t) + b_i^{(l)}\right)\right\} \\
&\qquad \odot \phi\left(W_c^{(l)}x_2^{(l)}(t) + U_c^{(l)}h_2^{(l)}(t) + b_c^{(l)}\right),
\end{aligned}$$

and

$$\begin{aligned}
&h_1^{(l)}(t+1) - h_2^{(l)}(t+1) \\
&= \sigma\left(W_o^{(l)}x_1^{(l)}(t) + U_o^{(l)}h_1^{(l)}(t) + b_o^{(l)}\right) \odot \left\{\phi\left(c_1^{(l)}(t+1)\right) - \phi\left(c_2^{(l)}(t+1)\right)\right\} \\
&\quad + \left\{\sigma\left(W_o^{(l)}x_1^{(l)}(t) + U_o^{(l)}h_1^{(l)}(t) + b_o^{(l)}\right) - \sigma\left(W_o^{(l)}x_2^{(l)}(t) + U_o^{(l)}h_2^{(l)}(t) + b_o^{(l)}\right)\right\} \odot \phi\left(c_2^{(l)}(t+1)\right).
\end{aligned}$$

Since the Lipschitz constant of $\sigma$ is $1/4$, we obtain that

$$
\left\| c_1^{(l)}(t+1) - c_2^{(l)}(t+1) \right\|
$$
$$
\leqslant \left\| \sigma\left( W_f^{(l)} x_1^{(l)}(t) + U_f^{(l)} h_1^{(l)}(t) + b_f^{(l)} \right) \right\|_\infty \left\| c_1^{(l)}(t) - c_2^{(l)}(t) \right\|
$$
$$
+ \frac{1}{4} \left\| \left( W_f^{(l)} x_1^{(l)}(t) + U_f^{(l)} h_1^{(l)}(t) + b_f^{(l)} \right) - \left( W_f^{(l)} x_2^{(l)}(t) + U_f^{(l)} h_2^{(l)}(t) + b_f^{(l)} \right) \right\| \left\| c_2^{(l)}(t) \right\|_\infty
$$
$$
+ \left\| \sigma\left( W_i^{(l)} x_1^{(l)}(t) + U_i^{(l)} h_1^{(l)}(t) + b_i^{(l)} \right) \right\|_\infty \left\| \left( W_c^{(l)} x_1^{(l)}(t) + U_c^{(l)} h_1^{(l)}(t) + b_c^{(l)} \right) \right.
$$
$$
\left. - \left( W_c^{(l)} x_2^{(l)}(t) + U_c^{(l)} h_2^{(l)}(t) + b_c^{(l)} \right) \right\|
$$
$$
+ \frac{1}{4} \left\| \left( W_i^{(l)} x_1^{(l)}(t) + U_i^{(l)} h_1^{(l)}(t) + b_i^{(l)} \right) \right.
$$
$$
\left. - \left( W_i^{(l)} x_2^{(l)}(t) + U_i^{(l)} h_2^{(l)}(t) + b_i^{(l)} \right) \right\| \left\| \phi\left( W_c^{(l)} x_2^{(l)}(t) + U_c^{(l)} h_2^{(l)}(t) + b_c^{(l)} \right) \right\|_\infty
$$
$$
\leqslant \overline{\sigma}_f^{(l)}(k) \left\| c_1^{(l)}(t) - c_2^{(l)}(t) \right\| + \frac{1}{4} \left( \left\| W_f^{(l)} \right\| \left\| x_1^{(l)}(t) - x_2^{(l)}(t) \right\| + \left\| U_f^{(l)} \right\| \left\| h_1^{(l)}(t) - h_2^{(l)}(t) \right\| \right) \overline{c}^{(l)}(k)
$$
$$
+ \overline{\sigma}_i^{(l)}(k) \left( \left\| W_c^{(l)} \right\| \left\| x_1^{(l)}(t) - x_2^{(l)}(t) \right\| + \left\| U_c^{(l)} \right\| \left\| h_1^{(l)}(t) - h_2^{(l)}(t) \right\| \right)
$$
$$
+ \frac{1}{4} \overline{\phi}_c^{(l)}(k) \left( \left\| W_i^{(l)} \right\| \left\| x_1^{(l)}(t) - x_2^{(l)}(t) \right\| + \left\| U_i^{(l)} \right\| \left\| h_1^{(l)}(t) - h_2^{(l)}(t) \right\| \right),
$$

and

$$
\left\| h_1^{(l)}(t+1) - h_2^{(l)}(t+1) \right\|
$$
$$
\leqslant \overline{\sigma}_o^{(l)}(k) \left\| c_1^{(l)}(t+1) - c_2^{(l)}(t+1) \right\|
$$
$$
+ \frac{1}{4} \phi\left( \overline{c}^{(l)}(k) \right) \left( \left\| W_o^{(l)} \right\| \left\| x_1^{(l)}(t) - x_2^{(l)}(t) \right\| + \left\| U_o^{(l)} \right\| \left\| h_1^{(l)}(t) - h_2^{(l)}(t) \right\| \right)
$$
$$
\leqslant \overline{\sigma}_o^{(l)}(k) \Big\{ \overline{\sigma}_f^{(l)}(k) \left\| c_1^{(l)}(t) - c_2^{(l)}(t) \right\|
$$
$$
+ \frac{1}{4} \left( \left\| W_f^{(l)} \right\| \left\| x_1^{(l)}(t) - x_2^{(l)}(t) \right\| + \left\| U_f^{(l)} \right\| \left\| h_1^{(l)}(t) - h_2^{(l)}(t) \right\| \right) \overline{c}^{(l)}(k)
$$
$$
+ \overline{\sigma}_i^{(l)}(k) \left( \left\| W_c^{(l)} \right\| \left\| x_1^{(l)}(t) - x_2^{(l)}(t) \right\| + \left\| U_c^{(l)} \right\| \left\| h_1^{(l)}(t) - h_2^{(l)}(t) \right\| \right)
$$
$$
+ \frac{1}{4} \overline{\phi}_c^{(l)}(k) \left( \left\| W_i^{(l)} \right\| \left\| x_1^{(l)}(t) - x_2^{(l)}(t) \right\| + \left\| U_i^{(l)} \right\| \left\| h_1^{(l)}(t) - h_2^{(l)}(t) \right\| \right) \Big\}
$$
$$
+ \frac{1}{4} \phi\left( \overline{c}^{(l)}(k) \right) \left( \left\| W_o^{(l)} \right\| \left\| x_1^{(l)}(t) - x_2^{(l)}(t) \right\| + \left\| U_o^{(l)} \right\| \left\| h_1^{(l)}(t) - h_2^{(l)}(t) \right\| \right).
$$

Then,

$$
\begin{pmatrix} \left\| c_1^{(l)}(t+1) - c_2^{(l)}(t+1) \right\| \\ \left\| h_1^{(l)}(t+1) - h_2^{(l)}(t+1) \right\| \end{pmatrix}
$$
$$
\leqslant \begin{bmatrix} \overline{\sigma}_f^{(l)}(k) & \alpha_s^{(l)}(k) \\ \overline{\sigma}_o^{(l)}(k)\overline{\sigma}_f^{(l)} & \alpha_s^{(l)}(k)\overline{\sigma}_o^{(l)}(k) + \frac{1}{4}\phi\left(\overline{c}^{(l)}(k)\right) \left\| U_o^{(l)} \right\| \end{bmatrix} \begin{pmatrix} \left\| c_1^{(l)}(t) - c_2^{(l)}(t) \right\| \\ \left\| h_1^{(l)}(t) - h_2^{(l)}(t) \right\| \end{pmatrix}
$$
$$
+ \begin{pmatrix} \alpha_x^{(l)}(k) \\ \alpha_x^{(l)}(k)\overline{\sigma}_o^{(l)}(k) + \frac{1}{4}\phi\left(\overline{c}^{(l)}(k)\right) \left\| W_o^{(l)} \right\| \end{pmatrix} \left\| x_1^{(l)}(t) - x_2^{(l)}(t) \right\|
$$
$$
=: A_s^{(l)}(k) \begin{pmatrix} \left\| c_1^{(l)}(t) - c_2^{(l)}(t) \right\| \\ \left\| h_1^{(l)}(t) - h_2^{(l)}(t) \right\| \end{pmatrix} + a_x^{(l)}(k) \left\| x_1^{(l)}(t) - x_2^{(l)}(t) \right\|,
$$

where $\alpha_s^{(l)}(k) = \frac{1}{4}\overline{c}^{(l)}(k)\|U_f^{(l)}\| + \overline{\sigma}_i^{(l)}(k)\|U_c^{(l)}\| + \frac{1}{4}\overline{\phi}_c^{(l)}(k)\|U_i^{(l)}\|$ and $\alpha_x^{(l)}(k) = \frac{1}{4}\overline{c}^{(l)}(k)\|W_f^{(l)}\| + \overline{\sigma}_i^{(l)}(k)\|W_c^{(l)}\| + \frac{1}{4}\overline{\phi}_c^{(l)}(k)\|W_i^{(l)}\|$.

Similar to the proof of Theorem A.4, we have

$$\|s_1(t) - s_2(t)\| \leqslant \mu\lambda_1(A_s(k))^t \|s_1(0) - s_2(0)\| + \|(I - A_s(k)^t)(I - A_s(k))^{-1}a_x(k)\| \|x_1 - x_2\|_{2,\infty}$$

for single-layer LSTM, and

$$
\begin{aligned}
&\left\|s_1^{(L)}(t) - s_2^{(L)}(t)\right\| \\
&\leqslant \mu^{(L)}\rho(A_s^{(L)}(k))^t \left\|s_1^{(L)}(0) - s_2^{(L)}(0)\right\| \\
&\quad + \sum_{m=1}^{L-1} \mu^{(m:L)} \frac{(t+L-m)!}{t!(L-m)!} \max_{l=m,\ldots,L} \rho(A_s^{(l)}(k))^t \left(\prod_{l=m+1}^{L} \left\|a_x^{(l)}(k)\right\|\right) \left\|s_1^{(m)}(0) - s_2^{(m)}(0)\right\| \\
&\quad + \sum_{\tau=0}^{t-1} \mu^{(1:L)} \frac{(t+L-\tau-2)!}{(t-\tau-1)!(L-1)!} \max_{l=1,\ldots,L} \rho(A_s^{(l)}(k))^{t-\tau-1} \left(\prod_{l=1}^{L} \left\|a_x^{(l)}(k)\right\|\right) \|x_1(\tau) - x_2(\tau)\|
\end{aligned}
$$

for multi-layer LSTM. Thus, LSTM is $\delta$ISS if $\rho(A_s^{(l)}(k)) < 1$ for all $l \in [L]$.

The proof of the monotonically decreasing of $\rho(A_s^{(l)}(k))$ is the same as that of Theorem A.4. $\qquad\square$

### A.3   Proof of Theorem 5.4

**Theorem A.6** (Upper Bound of Recovery Time)**.** *We assume that $s^{(l)}(0) \in \mathcal{S}^{(l)}(k)$ and $\rho(A_s^{(l)}(k)) < 1$ for all $l \in [L]$. Let infinite length inputs $x(t), \widehat{x}(t) \in [-x_{\max}, x_{\max}]^{n_x}, t \geqslant 0$ and $e > 0$. We assume that there exists $t_0$ such that $x(t) = \widehat{x}(t)$ for any $t \geqslant t_0$. Then, we have*

$$\sup_{s(0)\in\prod_{l=1}^{L} S_k^{(l)}} T_R(x, \widehat{x}, s(0); e) \leqslant \min\{t \geqslant 0 : \tilde{\beta}(2\sqrt{n_c}\bar{s}(k), t'; k) \leqslant e/\|U_y\| \text{ for any } t' \geqslant t\},$$

*where $\bar{s}(k) = (\bar{s}^{(1)}(k), \ldots, \bar{s}^{(L)}(k))$ and $\bar{s}^{(l)}(k) = \|(\bar{c}^{(l)}(k), \phi(\bar{c}^{(l)}(k))\bar{\sigma}_o^{(l)}(k))\|$.*

*Proof.* Denote that

$$
s_1^{(l)} = \begin{pmatrix} c_1^{(l)} \\ h_1^{(l)} \end{pmatrix} := s^{(l)}(t_0, s(0), x(0:t_0)),
$$

$$
s_2^{(l)} = \begin{pmatrix} c_2^{(l)} \\ h_2^{(l)} \end{pmatrix} := s^{(l)}(t_0, s(0), \widehat{x}(0:t_0))
$$

for $l = 1, \ldots, L$. Proposition 5.3 implies that there exists $\tilde{\beta}$ such that

$$
\begin{aligned}
&\|s^{(L)}(t - t_0, s_1(t_0), x(t_0:t)) - s^{(L)}(t - t_0, s_2(t_0), x(t_0:t))\| \\
&\leqslant \tilde{\beta}(\|s_1^{(1)} - s_2^{(1)}\|, \ldots, \|s_1^{(L)} - s_2^{(L)}\|, t - t_0; k)
\end{aligned}
$$

and $\tilde{\beta}(s^{(1)}, \ldots, s^{(L)}, t; k)$ is increasing with respect to $s^{(1)}, \ldots, s^{(L)}$ for any $t$ and $k$. On the other hand, $s_1^{(l)}, s_2^{(l)} \in \mathcal{S}^{(l)}(k)$ implies

$$
\begin{aligned}
\|c_1^{(l)} - c_2^{(l)}\| &\leqslant \sqrt{n_c}\|c_1^{(l)} - c_2^{(l)}\|_\infty \leqslant 2\sqrt{n_c}\bar{c}^{(l)}(k), \\
\|h_1^{(l)} - h_2^{(l)}\| &\leqslant \sqrt{n_c}\|h_1^{(l)} - h_2^{(l)}\|_\infty \leqslant 2\sqrt{n_c}\phi(\bar{c}^{(l)}(k))\bar{\sigma}_o^{(l)}(k),
\end{aligned}
$$

and then

$$\|s_1^{(l)} - s_2^{(l)}\| = \sqrt{\|c_1^{(l)} - c_2^{(l)}\|^2 + \|h_1^{(l)} - h_2^{(l)}\|^2} \leqslant 2\sqrt{n_c}\bar{s}^{(l)}(k).$$

Thus, we have

$$\|y(t, s(0), x(0 : t) - y(t, s(0), \widehat{x}(0 : t))\|$$
$$\leqslant \|U_y\| \|s^{(L)}(t, s(0), x(0 : t)) - s^{(L)}(t, s(0), \widehat{x}(0 : t))\|$$
$$= \|U_y\| \|s^{(L)}(t - t_0, s_1, x(t_0 : t)) - s^{(L)}(t - t_0, s_2, x(t_0 : t))\|$$
$$\leqslant \|U_y\| \tilde{\beta}(2\sqrt{n_c} \bar{s}(k), t - t_0; k).$$

The theorem follows immediately from this inequality. □

## B  Additional Discussion: Approximation of limit

As mentioned in Remark 4.3, if we assume the initial state as 0, the sufficient condition in Theorem 4.2 becomes optimal when $k \to \infty$, and also the estimate in Theorem 5.4 becomes more precise. Therefore, taking Theorem 4.2 as an example, calculating $\rho(A_s^{(l)}(\infty))$ or estimating the upper bound of it precisely is a meaningful goal. While iteratively calculating the values based on the recurrent formulas, which were used in the experiments in Section 7, is an approach for estimating the upper bound, we discuss another approach in this section. This approach may require more calculations than the iterative one, but it probably converges faster.

Below, we describe the method for calculating $\eta^{(l)}(\infty) \coloneqq \lim_{k \to \infty} \eta^{(l)}(k)$. The value of $\rho(A_s^{(l)}(\infty))$ and the limit as $k \to \infty$ of the variables in Theorem 5.4 can be directly calculated from $\eta^{(l)}(\infty)$.

Denote the limits of sequences $\{\overline{\sigma}_*^{(l)}(k)\}_{k=0}^{\infty}$ ($*$ denotes $f, i$, or $o$), $\{\overline{\phi}_c^{(l)}(k)\}_{k=0}^{\infty}$, and $\{\eta^{(l)}(k)\}_{k=-1}^{\infty}$ as $\overline{\sigma}_*^{(l)}(\infty)$, $\overline{\phi}_c^{(l)}(\infty)$ and $\eta^{(l)}(\infty)$, respectively. Since $\overline{\sigma}_*^{(l)}(\infty) = \sigma(G_*^{(l)}(\eta^{(l)}(\infty)))$ and $\overline{\phi}_c^{(l)}(\infty) = \phi(G_c^{(l)}(\eta^{(l)}(\infty)))$, we have

$$\eta^{(l)}(\infty) = \phi \left( \frac{\sigma(G_i^{(l)}(\eta^{(l)}(\infty)))\phi(G_c^{(l)}(\eta^{(l)}(\infty)))}{1 - \sigma(G_f^{(l)}(\eta^{(l)}(\infty)))} \right) \sigma(G_o^{(l)}(\eta^{(l)}(\infty))).$$

If this equation could be solved, that is, $\overline{\sigma}_*^{(l)}(\infty)$ and $\overline{\phi}_c^{(l)}(\infty)$ could be obtained, we would find the loosest sufficient condition for the LSTM to be $\delta$ISS, and the optimal estimate of recovery time. However, it is difficult to solve it analytically, so we approximate it using properties of convex functions.

Let $w \geqslant 0$. Since $\sigma(w)$ is convex upward and $\frac{d\sigma(w)}{dw} = \sigma(w)(1 - \sigma(w))$, then for $w$ and $w_0 \geqslant 0$,

$$\sigma(w) \leqslant \sigma(w_0) + \sigma(w_0)(1 - \sigma(w_0))(w - w_0) =: \breve{\sigma}(w; w_0).$$

As $\frac{d\phi(w)}{dw} = 1 - \phi^2(w)$, it holds that

$$\phi(w) \leqslant \phi(w_0) + (1 - \phi^2(w_0))(w - w_0) =: \breve{\phi}(w; w_0).$$

Therefore, for $w$ and $w_0 \geqslant 0$, we have

$$\sigma(G_*^{(l)}(w)) \leqslant \breve{\sigma}(G_*^{(l)}(w); G_*^{(l)}(w_0)),$$
$$\phi(G_c^{(l)}(w)) \leqslant \breve{\phi}(G_c^{(l)}(w); G_c^{(l)}(w_0)).$$

From these inequalities, the following inequality holds:

$$\sigma(G_o^{(l)}(w))\phi \left( \frac{\sigma(G_i^{(l)}(w))\phi(G_c^{(l)}(w))}{1 - \sigma(G_f^{(l)}(w))} \right) =: \overline{g}^{(l)}(w)$$

$$\leqslant \breve{\sigma}(G_o^{(l)}(w); G_o^{(l)}(w_0)) \breve{\phi} \left( \frac{\breve{\sigma}(G_i^{(l)}(w); G_i^{(l)}(w_0))\breve{\phi}(G_c^{(l)}(w); G_c^{(l)}(w_0))}{1 - \sigma(G_f^{(l)}(w))}; \frac{\sigma(G_i^{(l)}(w_0))\phi(G_c^{(l)}(w_0))}{1 - \sigma(G_f^{(l)}(w_0))} \right)$$

$$=: \breve{g}^{(l)}(w; w_0).$$

Here, $G_*(w)$, $\breve{\sigma}(w, w_0)$, $\breve{\phi}(w, w_0)$, and $(1 - \sigma(w))^{-1}$ are non-negative, monotonically increasing, and convex downward functions. Since $\breve{g}^{(l)}(w; w_0)$ is represented by the composition and product of these functions, it is also convex downward. Hence, for $0 \leqslant \kappa^{(l)} \leqslant \eta^{(l)}$, $0 \leqslant \lambda^{(l)} \leqslant 1$, and $\eta^{(l)+} := (1 - \lambda)\kappa^{(l)} + \lambda\eta^{(l)}$, it follows that

$$\overline{g}^{(l)}(\eta^{(l)+}) \leqslant \breve{g}^{(l)}(\eta^{(l)+}; \eta^{(l)}) \leqslant (1 - \lambda^{(l)})\breve{g}^{(l)}(\kappa^{(l)}; \eta^{(l)}) + \lambda^{(l)}\overline{g}^{(l)}(\eta^{(l)}).$$

In particular, if $\eta^{(l)}(\infty)$ holds $\overline{g}^{(l)}(\eta^{(l)}(\infty)) = \eta^{(l)}(\infty)$ and $\kappa^{(l)} \leqslant \eta^{(l)}(\infty) \leqslant \eta^{(l)}$, there exists $\lambda^{(l)}(\infty)$ such that $\eta^{(l)}(\infty) = (1 - \lambda^{(l)}(\infty))\kappa^{(l)} + \lambda^{(l)}(\infty)\eta^{(l)}$. In this case, we have the following:

$$(1 - \lambda^{(l)}(\infty))\kappa^{(l)} + \lambda^{(l)}(\infty)\eta^{(l)} \leqslant (1 - \lambda^{(l)}(\infty))\breve{g}^{(l)}(\kappa^{(l)}; \eta^{(l)}) + \lambda^{(l)}(\infty)\overline{g}^{(l)}(\eta^{(l)}),$$

$$\lambda^{(l)}(\infty) \leqslant \frac{\breve{g}^{(l)}(\kappa^{(l)}; \eta^{(l)}) - \kappa^{(l)}}{\eta^{(l)} - \kappa^{(l)} - \overline{g}^{(l)}(\eta^{(l)}) + \breve{g}^{(l)}(\kappa^{(l)}; \eta^{(l)})},$$

$$\eta^{(l)}(\infty) \leqslant \frac{\eta^{(l)} \breve{g}^{(l)}(\kappa^{(l)}; \eta^{(l)}) - \kappa^{(l)} \overline{g}^{(l)}(\eta^{(l)})}{\eta^{(l)} - \kappa^{(l)} - \overline{g}^{(l)}(\eta^{(l)}) + \breve{g}^{(l)}(\kappa^{(l)}; \eta^{(l)})} = \eta^{(l)} - \frac{(\eta^{(l)} - \kappa^{(l)})(\eta^{(l)} - \overline{g}^{(l)}(\eta^{(l)}))}{\eta^{(l)} - \kappa^{(l)} - \overline{g}^{(l)}(\eta^{(l)}) + \breve{g}^{(l)}(\kappa^{(l)}; \eta^{(l)})}.$$

Therefore, we can approximate $\eta^{(l)}(\infty)$ by following three steps:

1. Take $\kappa^{(l)}$ and $\eta^{(l)}$ such that $\kappa^{(l)} \leqslant \eta^{(l)}(\infty) \leqslant \eta^{(l)}$, for example, $\kappa^{(l)} = \overline{g}^{(l)}(0)$ and $\eta^{(l)} = \overline{g}^{(l)}(1)$.

2. Calculate $\dfrac{\eta^{(l)} \breve{g}(\kappa^{(l)}; \eta^{(l)}) - \kappa^{(l)} \overline{g}^{(l)}(\eta^{(l)})}{\eta^{(l)} - \kappa^{(l)} - \overline{g}^{(l)}(\eta^{(l)}) + \breve{g}^{(l)}(\kappa^{(l)}; \eta^{(l)})}$.

3. Substitute the result of Step 2 for $\eta^{(l)}$. Return to Step 2.

## C  Details of Experimental Setup

This section provides a detailed description of the experimental settings used in the experiments discussed in the main text. The experiments were conducted using the PyTorch framework. The random seed was set using `torch.manual_seed(1234)`.

### C.1  Simplified Model

### C.1.1  LSTM Setup

The model dimensions are provided in Table 2. Twenty models were generated with weights sampled uniformly from the intervals $W_* \in [0, 0.1]$, $U_* \in [0, 1.0]$, and $C \in [0, 1.0]$.

| $L$ | $n_x$ | $n_c$ | $n_y$ |
|-----|-------|-------|-------|
| 1   | 1     | 1     | 1     |

Table 2: Model dimensions of simplified LSTM

### C.1.2  Evaluation

We define the nominal input data $x$ and the perturbed input data $\hat{x}$ that satisfy the assumptions of Theorem 5.4 with $t_0 = 20$ as follows.

$$x(t) = 0.5, \ \hat{x}(t) = \begin{cases} 1.0 & \text{if } t = 20, \\ 0.5 & \text{else}, \end{cases}$$

where $t = 0, 1, 2, \ldots, 99$. $T_R(x, \hat{x})$ and $\bar{T}_R(k)$ were calculated according to Algorithm 1 and Algorithm 2 where $\mathcal{T} = 100$. It should be noted that in cases where the model exhibits instability or $\rho(A_s(k)) \geqslant 1$, this algorithm computes $T_R = \infty$ or $\bar{T}_R(k) = \infty$. Calculating statistical measures such as estimation errors or

correlation coefficients using infinite values would result in arbitrary and meaningless numbers. Indeed, for some values of $k$ and specific models, our experiments showed that $\bar{T}_R(k) = \infty$ and $\rho(A_s(k)) \geqslant 1$ occurred. This result is theoretically correct and does not indicate any deficiency in the theory. However, using infinite values to calculate statistical measures such as estimation errors or correlation coefficients would result in arbitrary and meaningless numbers. To address this issue, we replaced infinity values with finite proxies: $\bar{T}_R(k) = \mathcal{T} = 100$ when computing these statistical measures.

---

**Algorithm 1** Computation of $T_R(x, \hat{x})$

---

1: $T \leftarrow \text{length}(x)$
2: $d \leftarrow \{|y(t, s(0), x(0:t)) - y(t, s(0), \hat{x}(0:t))|\}_{t=0,1,\dots,T}$
3: **if** $d[-0] < e$ **then**
4:     $T_R \leftarrow T - T_0$
5: **else**
6:     **return** $\infty$
7: **end if**
8: **for** $t = 1$ **to** $T - t_0$ **do**
9:     **if** $d[-t] < e$ **then**
10:       $T_R \leftarrow T_R - 1$
11:     **else**
12:       **return** $T_R - t_0$
13:     **end if**
14: **end for**
15: **return** $0$

---

**Algorithm 2** Computation of $\bar{T}_R(k)$

---

1: **Inputs:** $\mathcal{T}$
2: $d \leftarrow \{\tilde{\beta}(2\sqrt{n_c}\bar{s}(k), t; k)\}_{t=0,\dots,T}$
3: **if** $d[-0] < e$ **then**
4:     $\bar{T}_R \leftarrow \mathcal{T}$
5: **else**
6:     **return** $\infty$
7: **end if**
8: **for** $t = 1$ **to** $\mathcal{T}$ **do**
9:     **if** $d[-t] < e$ **then**
10:       $\bar{T}_R \leftarrow \bar{T}_R - 1$
11:     **else**
12:       **return** $\bar{T}_R$
13:     **end if**
14: **end for**
15: **return** $0$

---

## C.2 Two Tank System

### C.2.1 LSTM Setup

The model dimensions are provided in Table 3.

| $L$ | $n_x$ | $n_c$ | $n_y$ |
|---|---|---|---|
| 1 | 3 | 22 | 2 |

Table 3: Model dimensions of the two-tank system LSTM

### C.2.2 Data Generation

The training and test data were randomly generated through the following process (Jung et al., 2023; Schoukens & Noël, 2017). First, the continuous two-tank system

$$\frac{dh_1}{dt} = -p_1\sqrt{2gh_2} + p_2 u,$$
$$\frac{dh_2}{dt} = p_3\sqrt{2gh_1} - p_4\sqrt{2gh_2},$$

was discretized using the Runge-Kutta method with $dt = 0.01$. The system parameters are shown in Table 4. We randomly generated a sequence of control inputs $\{u(t)\}_t$, set the initial state of the system to $(h_1(0), h_2(0)) = (0, 0)$, and obtained the system states $\{(h_1(t), h_2(t))\}_t$. Here, the generation of control inputs was set to a time series length of $100,000$, with values randomly switched every $4,000$ step following a uniform distribution $\mathcal{U}(0.5, 3.0)$. Using this method, we created two independent control input sequences, one for training and one for testing.

As the final stage of the data generation process, we created training and test datasets. In this step, we simulated more realistic scenarios by adding noise to the previously generated state variables. Specifically, we constructed the input data as follows:

$$x(t) = (u(t), h_1(t) + w_1(t), h_2(t) + w_2(t)).$$

This addition of noise simulates real-world factors such as sensor measurement errors and environmental uncertainties. The noise distribution was set to $w_1, w_2 \sim \mathcal{N}(0, 0.1^2)$ for the training data and to $w_1, w_2 \sim \mathcal{N}(0, 0.01^2)$ for the test data. This is because while noise with a standard deviation of about 0.01 is expected under operational conditions, intentionally introducing larger noise during the training phase aims to improve the model's robustness and accuracy. On the other hand, the inference target data was set using the state of the next time step as follows:

$$y(t) = (h_1(t+1), h_2(t+1)).$$

This setup requires the model to predict the state at the next time step based on the current control input and the state, which includes noise. This simulates the important task of one-step-ahead state prediction in actual system control.

The time series data are normalized to the interval $[-1, 1]$ for both input $x(t) = (u(t), h_1(t) + w(t), h_2(t) + w(t))$ and output $y(t) = (h_1(t+1), h_2(t+1))$ of the LSTM before conducting inference. The equation for normalization is given below:

$$x = 2(x - x_{\min})/(x_{\max} - x_{\min}) - 1,$$
$$y = 2(y - y_{\min})/(y_{\max} - y_{\min}) - 1,$$

where $x_{\min} = (0.5, 0, 0)$, $x_{\max} = (3.0, 10, 10)$, $y_{\min} = (0, 0)$, and $y_{\max} = (10, 10)$.

| System constants | $p_1$ | $p_2$ | $p_3$ | $p_4$ | $g$ |
|---|---|---|---|---|---|
| | 0.5 | 0.5 | 0.5 | 0.5 | 0.5 |
| **Actuator** | $u_{\min}$ | $u_{\max}$ | | | |
| | 0.5 | 3.0 | | | |

Table 4: System parameters

### C.2.3 Train

**Data set** The training dataset is split into training and validation sets, with the first 50% of the sequences allocated for training and the remaining 50% used for validation purposes. To reduce computational load and

augment data, the training dataset is created by segmenting one length time series $50,000$ into individual sequences $49,000$, each with a length of $1,000$, using a sliding window of step size 1. On the other hand, as data augmentation is not required for the validation dataset, we create the dataset using both a window length and a step size of $1,000$, solely considering computational efficiency.

**Loss** We design the loss function for backward as follows:

$$\mathcal{L} = \text{MAE}(y_{\text{true}}, y_{\text{pred}}) + \lambda \Phi(\varepsilon).$$

When calculating the MAE on the batched data, we discard the first 10 steps of the time series predictions. This adjustment accounts for the inherent instability of LSTM predictions during the initial time steps, which is a characteristic behavior of LSTM networks.

**Hyperparameters** We used the following hyperparameters. The weights were initialized with uniform distribution $\mathcal{U}(-1/\sqrt{n_c}, 1/\sqrt{n_c})$ and optimized using the Adam ($\beta = (0.9, 0.999)$, $\varepsilon = 10^{-8}$) optimizer. The mini-batch size was 64, the epoch size was 200, the learning rate was 0.001 with ReduceLROnPlateau (mode = min, factor = 0.1, patience = 10) scheduler. The baseline model was constructed with $\lambda = 0$, while the stability-guaranteed model was implemented with $\lambda = 1.0$. The recovery time control parameter $\varepsilon$ was selected from the following set

$$\{0.1, 0.15, 0.2, 0.25, 0.3, 0.35, 0.4, 0.45, 0.5\}.$$

### C.2.4 Evaluation

The inference accuracy was evaluated using MAE as the metric for the test data. For the time series, we calculated the MAE, ignoring the first 10 steps and using non-normalized values for the evaluation.

Separate from the data used for evaluating inference accuracy, we generate the nominal input data $x$ and the perturbed input data $\hat{x}$ for the estimation of recovery time $T_R$ using the following steps. First, we set the time series length to 3500 steps, the control input to $u = 1.0$, and the initial water level of the system to $(h_1(0), h_2(0)) = (0, 0)$. Then, following the data generation process, we created $(h_1(t), h_2(t))$ and set $x(t) = (u(t), h_1(t), h_2(t))$. Then, we define $\hat{x}$ that satisfies the assumptions of Theorem 5.4 with $t_0 = 1010$ as

$$\hat{x}(t) = \begin{cases} (u(t), h_1(t) + 1.0, h_2(t) + 1.0) & \text{if } 1000 \leqslant t \leqslant 1010, \\ x(t) & \text{else.} \end{cases}$$

This definition models the phenomenon where observational noise temporarily has a magnitude larger than expected. Based on these settings, we calculate $T_R$ and $\bar{T}_R$. The algorithm is similar to the one used with the toy model, but there is one point to note due to the effect of data normalization. As the LSTM output is normalized, $T_R$ is calculated to ensure the error remains within e on the original scale. Consequently, $\bar{T}_R$ must also be adjusted to the original scale. In the calculation of $\bar{T}_R$, we apply the following scale transformation to $e$ in Theorem 5.4:

$$e \leftarrow e/(h_{\max} - h_{\min}) = e/(10 - 0) = e/10.$$

## D Other Related Work

In this paper, we estimated recovery time using $\delta$ISS properties. On the other hand, Finite time stability (FTS) (Bhat & Bernstein, 2000) rigorously discusses convergence time, characterized by a settling-time function. Finite-time input-to-state stability (FTISS), combining FTS and ISS, enables robust control against disturbances (He et al., 2022). Recent studies (Liang & Liang, 2023; Mo et al., 2023) propose Lyapunov functions satisfying FTISS, eliminating the need to directly consider the settling-time function.

However, deriving sufficient FTS conditions involves strict model constraints, established only for specific models like Hopfield networks (Michalak & Nowakowski, 2017) and Clifford-valued RNNs (Shen & Meng,

2023), but not for conventional RNNs or LSTMs. While achieving FTS for LSTMs would theoretically allow for precise recovery time evaluation, we argue that from a practical standpoint, considering numerical errors and rounding, it suffices for output differences to fall below a threshold rather than strictly reaching equilibrium. Thus, we opted for $\delta$ISS instead of FTS.

