# OpenReview forum: "Quantification and Control of LSTM Resilience Based on Stability Theory"
_TMLR — Decision pending for TMLR_

### Review · Reviewer_eMob · 2026-04-04

**Summary Of Contributions:**

The authors present a theoretical framework to guarantee and evaluate the resilience of Long Short-Term Memory (LSTM) networks used in control systems. The paper introduces a new metric called recovery time which measures the time needed for an LSTM unit to return to its normal operation state after it experiences a disturbance or receives unexpected input. The authors developed a data-independent recovery time upper limit through their work which uses an iterative relaxation parameter to establish the parameter's value. The authors use the upper limit to develop a training method which measures system resilience through a loss penalty that maintains control over state transition spectral radius limits. The theoretical claims are validated empirically on a simplified synthetic model and a classical two-tank control benchmark which shows a distinct relationship between inference accuracy and model resilience.

Key Strengths:
- The mathematical formalization of the invariant sets and the relaxation of the $\delta$ISS conditions (Theorem 4.2) are elegant and mathematically sound.
- Providing a computable, data-independent upper bound for recovery time allows for a priori safety guarantees, which is highly valuable for safety-critical control systems.
- Translating the theoretical bounds into a differentiable loss penalty allows practitioners to directly tune the stability-plasticity trade-off during training.

Key Weaknesses:
- The iterative calculation of the invariant sets up to step $k$ during each training step implies a computational overhead that is not fully quantified.
- As is common with Lipschitz-based stability bounds in neural networks, the upper bound on recovery time can be quite conservative, potentially over-penalizing the network and degrading performance on tasks requiring long-term memory dependencies.

**Additional Comments:**

The mathematical formatting and notation are exceptionally clean and well-structured, making a dense paper surprisingly readable. I commend the authors for their thoroughness in the appendix proofs.

**Audience:**

Yes

**Audience Explanation:**

The field of control-theoretic safety guarantees and deep learning system validation requires continuous research to establish safety standards. The formal definition of recovery time together with the method to restrict spectral radius of LSTM dynamics will benefit researchers who work on Model Predictive Control and safety-critical AI systems and robust time-series forecasting.

**Broader Impact Concerns:**

I do not have any major ethical concerns.

**Claims And Evidence:**

Yes

**Claims Explanation:**

The mathematical claims are supported by proofs which exist in the appendix. The process generates two results because it derives the contraction invariant sets which prove that spectral radius $\rho(A_s^{(l)}(k))$ decreases when k increases. The experiments provide evidence which confirms the theoretical results. The first experiment shows that increasing k will make recovery time limits more accurate while it will also move more randomly initialized models into the guaranteed delta ISS area. The Two-Tank System Experiment II demonstrates how the loss penalty works in real life by showing a Pareto frontier trade-off between Mean Absolute Error MAE and recovery time as the penalty parameter epsilon changes.

**Requested Changes:**

I recommend the following adjustments to strengthen the manuscript prior to acceptance.

Critical for Acceptance:
- The proposed resilience-aware training requires computing the recurrent formulas for the invariant sets up to step $k$ at each parameter update. The paper currently lacks a discussion on the computational complexity of this operation. The study needs a short empirical profiling of training overheads which should include time and memory requirements together with a Big-$\mathcal{O}$ analysis that shows how these overheads change when k increases from 0 to 20 and higher.
- The authors openly declare in their limitations section that tasks which depend on extensive long-term memory will experience severe problems because of the resilience penalty. The paper would achieve greater strength through empirical proof of the failure mode which the authors propose. The demonstration of $\epsilon$ penalty implementation which results in total loss of long-term knowledge would provide practitioners with better insight into the method's actual operational boundaries.

To Strengthen the Work:
- The current comparison between adversarial training and data-driven training methods exists in Appendix D. Most deep learning practitioners use standard data augmentation as their basic method, so we need to move Table 4 with its analysis into the main text to create a more powerful study which shows the real worth of our theoretical guarantees between Section 7 and the main text.
- In Section 6.2, it is noted that $\Phi(\epsilon)$ approaches 0 as training progresses. A brief discussion on how often this convergence actually occurs in practice for complex, real-world datasets (vs. the simpler ones tested) would be beneficial, as forcing $\rho < 1$ globally is a very strong constraint.

---

> ### Author Response · Authors · 2026-04-29
> **Response to reviewer eMob part1**
>
> Thank you very much for your positive and encouraging review. We appreciate your recognition of the motivation behind our work. We will comment on your questions and concerns below:
>
> *---The proposed resilience-aware training requires computing the recurrent formulas for the invariant sets up to step at each parameter update. The paper currently lacks a discussion on the computational complexity of this operation. The study needs a short empirical profiling of training overheads which should include time and memory requirements together with a Big-$\mathcal{O}$ analysis that shows how these overheads change when k increases from 0 to 20 and higher.*
>
> To perform resilience-aware training, it is necessary to additionally calculate $\rho(A(k))$, so we evaluated the overhead of this calculation in a simple experiment. Specifically, for single-layer LSTMs with three different numbers of hidden units, we measured the time and memory usage required to calculate $\rho(A(k))$ in both forward propagation without gradients and backpropagation with gradients. From Tables 1-2, the training time is $O(k)$ in both the no-grad and the with grad. For memory, it can be evaluated as $O(1)$ because $\rho(A(k))$ can be calculated from the recurrence relation. This trend can be confirmed in Table 3. The slight increase in memory consumption when $k$ is increased in Table 4 is likely due to the calculation of the backward gradient. As you pointed out, we agree that this discussion is practically important, so we will add a brief comment to the main text and detailed experimental results to the appendix of the paper.
>
> Note added on May 3, 2026: All measurements reported in Tables 1–4 were conducted on a Google Colab T4 GPU.
>
> ---
>
> ## Table 1: Average Time (seconds) over 50 trials for calculation of $\rho(A(k))$ without gradients, for three LSTMs with different hidden sizes (n_hidden).
>
> | n_hidden \ k | 0 | 10 | 20 | 30 | 40 | 50 |
> |---|---|---|---|---|---|---|
> | 32 | 0.005353 | 0.009221 | 0.010908 | 0.015149 | 0.018689 | 0.023855 |
> | 64 | 0.008745 | 0.010760 | 0.020215 | 0.022202 | 0.022830 | 0.032025 |
> | 128 | 0.015649 | 0.020495 | 0.022376 | 0.025963 | 0.036796 | 0.033801 |
>
> ---
>
> ## Table 2: Average Time (seconds) over 50 trials for calculation of $\rho(A(k))$ with gradients, for three LSTMs with different hidden sizes (n_hidden).
>
> | n_hidden \ k | 0 | 10 | 20 | 30 | 40 | 50 |
> |---|---|---|---|---|---|---|
> | 32 | 0.008097 | 0.022416 | 0.039670 | 0.054096 | 0.087476 | 0.091171 |
> | 64 | 0.010515 | 0.030754 | 0.046505 | 0.062178 | 0.101995 | 0.091318 |
> | 128 | 0.018488 | 0.038326 | 0.049763 | 0.069685 | 0.085944 | 0.116248 |
>
> ---
>
> ## Table 3: Average Memory (MB) over 50 trials for calculation of $\rho(A(k))$ without gradients, for three LSTMs with different hidden sizes (n_hidden).
>
> | n_hidden \ k | 0 | 10 | 20 | 30 | 40 | 50 |
> |---|---|---|---|---|---|---|
> | 32 | 9.130371 | 8.599121 | 8.599121 | 8.599121 | 8.599121 | 8.599121 |
> | 64 | 8.758301 | 8.789551 | 8.789551 | 8.789551 | 8.789551 | 8.789551 |
> | 128 | 9.318359 | 9.443359 | 9.443359 | 9.443359 | 9.443359 | 9.443359 |
>
> ---
>
> ## Table 4: Average Memory (MB) over 50 trials for calculation of $\rho(A(k))$ with gradients, for three LSTMs with different hidden sizes (n_hidden).
>
> | n_hidden \ k | 0 | 10 | 20 | 30 | 40 | 50 |
> |---|---|---|---|---|---|---|
> | 32 | 8.914902 | 9.216309 | 9.528809 | 9.841309 | 10.153809 | 10.466309 |
> | 64 | 9.158613 | 9.997559 | 10.837402 | 11.677246 | 12.517090 | 13.356934 |
> | 128 | 10.137324 | 12.966309 | 15.798340 | 18.630371 | 21.462402 | 24.294434 |

---

> ### Author Response · Authors · 2026-04-29
> **Response to reviewer eMob part2**
>
> *--The authors openly declare in their limitations section that tasks which depend on extensive long-term memory will experience severe problems because of the resilience penalty. The paper would achieve greater strength through empirical proof of the failure mode which the authors propose. The demonstration of penalty implementation which results in total loss of long-term knowledge would provide practitioners with better insight into the method's actual operational boundaries.*
>
> To measure long-term knowledge, we conducted an LSTM in the task of memorizing numbers. The LSTM was trained to output $x$ for every input that is $x$ for the first 10 times and $0$ thereafter. The best results were achieved without a resilience penalty (i.e., $\lambda=0$), with an MAE of nearly $0$, indicating perfect memorization. Conversely, long-term memory deteriorated when a resilience penalty was applied, but it's noteworthy that the MAE remains roughly the same regardless of the value of the regularization parameter $\lambda>0$.
>
> ---
>
> ## Average MSE (at step 2000) over 3 samples for Different initial input length (seq_len) and penalty coefficient lambda
>
> | seq_len \ lambda | 0.0 | 0.1 | 0.2 | 0.3 | 0.4 | 0.5 | 0.6 | 0.7 | 0.8 | 0.9 | 1.0 |
> |---|---|---|---|---|---|---|---|---|---|---|---|
> | 10 | 0.0000 | 0.0017 | 0.0023 | 0.0004 | 0.0025 | 0.0053 | 0.0017 | 0.0030 | 0.0041 | 0.0035 | 0.0038 |
> | 50 | 0.0006 | 0.0993 | 0.1194 | 0.0186 | 0.0357 | 0.1546 | 0.1074 | 0.0666 | 0.0349 | 0.0948 | 0.0788 |
> | 100 | 0.0005 | 0.1375 | 0.0845 | 0.1068 | 0.0780 | 0.0293 | 0.0338 | 0.1048 | 0.0754 | 0.0954 | 0.1351 |
>
> ---
>
> *--The current comparison between adversarial training and data-driven training methods exists in Appendix D. Most deep learning practitioners use standard data augmentation as their basic method, so we need to move Table 4 with its analysis into the main text to create a more powerful study which shows the real worth of our theoretical guarantees between Section 7 and the main text.*
>
> Thank you for the advice. We will move Section D between Section 7 and Section 8.

---

> ### Author Response · Authors · 2026-04-29
> **Response to reviewer eMob part3**
>
> *--In Section 6.2, it is noted that $\Phi(\epsilon)$ approaches 0 as training progresses. A brief discussion on how often this convergence actually occurs in practice for complex, real-world datasets (vs. the simpler ones tested) would be beneficial, as forcing globally is a very strong constraint.*
>
> We did not record the change in the value of $\Phi(\varepsilon)$ during training for the experiment in Section 7.2 because we determined that the trade-off could be confirmed by looking at the recovery time and MAE. However, it appears that $\Phi(\varepsilon)$ converged to 0 when $\lambda >0$. Besides, in all cases with $\lambda>0$, involving the above memorizing numbers experiment, $\rho(A)$ converged to less than $1$, thus $\Phi(\varepsilon)\to 0$.
>
> ---
>
> ## Average Spectral Radius (at step 2000) over 3 samples for Different initial input length (seq_len) and penalty coefficient lambda
>
> | seq_len \ lambda | 0.0 | 0.1 | 0.2 | 0.3 | 0.4 | 0.5 | 0.6 | 0.7 | 0.8 | 0.9 | 1.0 |
> |---|---|---|---|---|---|---|---|---|---|---|---|
> | 10 | 6.968×10² | 0.9953 | 0.9918 | 0.9803 | 0.9953 | 0.9792 | 0.9909 | 0.9819 | 0.9952 | 0.9871 | 0.9854 |
> | 50 | 1.752×10² | 0.9905 | 0.9795 | 0.9887 | 0.9923 | 0.9943 | 0.9818 | 0.9854 | 0.9823 | 0.9970 | 0.9912 |
> | 100 | 1.394×10² | 0.9910 | 0.9990 | 0.9923 | 0.9969 | 0.9927 | 0.9856 | 0.9961 | 0.9950 | 0.9966 | 0.9905 |

---

### Review · Reviewer_Y2nZ · 2026-04-19

**Summary Of Contributions:**

This paper gives finite-time stabilization guarantees for LSTM with a practical upper bound as main contribution. The paper is primarily theoretical in nature.

**Additional Comments:**

Motivation for paper was using LSTMs as controllers for dynamical systems. Now, you treat the LSTM as the dynamical system. What do your results imply for the dynamical system controlled by the LSTM? Is there real consequence for stability?

**Audience:**

Yes

**Audience Explanation:**

An interested readership may find curiosity in the paper. Due to the limited impact of upper bound (above) they may be disappointed.

The readership will also profit from an improved presentation. There are some typos. The use of stability to mean different things in the paper is confusing (page 1 vs page 3).

**Broader Impact Concerns:**

Using upper bounds without lower bounds as performance indicators is dangerous. It could invite less good research practice.

**Claims And Evidence:**

Yes

**Claims Explanation:**

While I was able to follow the proofs broadly, the general audience will profit from more explanations and shorter equations in the proofs.
I could not follow a step on page 20:
 - The bottom of page 20 shows that the mu^(l) term depends on t
 - but the first equation on page 20 treats it as a constant.
 - This could work with monotonicity of mu^(l) in tau_l, but I don't think this is given.

The datasets used are small. Can be acceptable for a theoretical work.

The main practical contribution is an upper bound on stabilization time. The goal to establish it as a standard comparison target is perhaps unadvisable as it can be misleading. Comparing with an upper bound does not reveal whether an LSTM just made the bound tighter or actually improved stabilization. A well-stabilizing LSTM can also have a loose upper bound.

**Requested Changes:**

Please improve the presentation. The proof should be easier to understand. Also answer my question above.

---

> ### Author Response · Authors · 2026-05-07
> **Response to reviewer Y2nZ**
>
> Thank you very much for your positive and encouraging review. We appreciate your recognition of the motivation behind our work. We will comment on your questions and concerns below:
>
> *--The bottom of page 20 shows that the mu^(l) term depends on t*
> We thank you for your careful review. We have corrected $\mu^{(l)}$ to $\mu^{(l)}(t)$ all of these errors in page 20.
>
> *--but the first equation on page 20 treats it as a constant.*
> *--This could work with monotonicity of mu^(l) in tau_l, but I don't think this is given.*
> The reason $M_L(k), M'_L(k)$, and $M''_L(k)$ in page 20 are constants is $\rho(A^{{l}}_s(k))<\rho(A^{{L}}_s(k))$ and the products of $\mu$ and $\rho(A^{{L}}_s(k))$ are converge.
>
> *--The main practical contribution is an upper bound on stabilization time. The goal to establish it as a standard comparison target is perhaps unadvisable as it can be misleading. Comparing with an upper bound does not reveal whether an LSTM just made the bound tighter or actually improved stabilization. A well-stabilizing LSTM can also have a loose upper bound.*
> We agree that there is a gap between the recovery upper bound of LSTM and the real stability of LSTM. The main purpose of this paper is to guarantee the recovery of that LSTM model by theoretically deriving an upper bound on the recovery time, and not to compare the stability of different models based on this. However, experimental results suggest that the upper-bound analysis is actually useful in controlling the model's recovery time.
>
> *--The readership will also profit from an improved presentation. There are some typos. The use of stability to mean different things in the paper is confusing (page 1 vs page 3).*
> There is some overlap between the literal meaning of stability and the meaning of stability in the context of δISS, but the latter is encompassed within the former.
>
> *--Using upper bounds without lower bounds as performance indicators is dangerous. It could invite less good research practice.*
> According to definition 5.1, if $\hat{x}(t)=x(t)$ for all $t$, then $T_R=0$. Therefore, the lower bound of recovery time is always $0$.
>
> *--Motivation for paper was using LSTMs as controllers for dynamical systems. Now, you treat the LSTM as the dynamical system. What do your results imply for the dynamical system controlled by the LSTM? Is there real consequence for stability?*
> As stated in the abstract, the motivation for this paper is to introduce and evaluate recovery time for guaranteeing and evaluating the resilience of LSTM. Furthermore, we introduced a method for controlling the resilience of LSTMs using this evaluation. While development for use as a system controller is conceivable, it falls outside the scope of this paper and is therefore not explored.

---

> > ### Comment · Reviewer_Y2nZ · 2026-05-14
> >
> > Unfortunately, I could not find the places on former page 20 that were supposed to be updated with $\mu^(l)(t)$. Perhaps those were not included in the posted revision?
> >
> > Similarly, I did not witness improvements to the presentation and I can still see several typos.
> >
> > The trivial lower bound $0$ on the recovery time is clear from the definition but not informative. As such, it does not address my concern on research quality impact from suggested practice of using the upper bound from the paper in isolation.
> >
> > First sentence of the abstract reads "This paper proposes a novel theoretical framework for guaranteeing and evaluating the resilience of long short-term memory (LSTM) networks **in control systems**". As such my question about LSTMs as controllers was warranted but not answered satisfactory.
> >
> > I appologize to AC for late response.

---

> > > ### Author Response · Authors · 2026-05-15
> > > **Second Response to Reviewer Y2nZ**
> > >
> > > We have revised the manuscript. We apologize for the delay. We would appreciate it if you could check.

---

> > > ### Author Response · Authors · 2026-05-16
> > > **Second Response to Reviewer Y2nZ (part2)**
> > >
> > > Thank you very much for your patience and for your continued critical review of our work. Following our recent submission of the revised manuscript, we have provided our point-by-point responses below to detail how we addressed your remaining concerns:
> > >
> > > *--The trivial lower bound $0$ on the recovery time is clear from the definition but not informative. As such, it does not address my concern on research quality impact from suggested practice of using the upper bound from the paper in isolation.*
> > >
> > > The upper bound guarantees that recovery will occur within a certain time, while the lower bound simply indicates that recovery will not occur within that time. If the difference between the upper and lower bounds can be made sufficiently small, it becomes possible to estimate the recovery time, but such a sharp evaluation is likely to be difficult. We do not recommend using the theoretical upper bound as a standalone performance indicator. In practice, this upper bound complements empirical recovery time estimation under perturbations/adversarial disturbances, and provides a theoretical guarantee of recovery.
> > >
> > > *--First sentence of the abstract reads "This paper proposes a novel theoretical framework for guaranteeing and evaluating the resilience of long short-term memory (LSTM) networks in control systems". As such my question about LSTMs as controllers was warranted but not answered satisfactory.*
> > >
> > > As the reviewer points out, we are discussing long short-term memory (LSTM) networks in control systems. However, this paper focuses on using them as observers (as in Section 7.2), rather than as controllers.

---

### Review · Reviewer_SD9P · 2026-05-04

**Summary Of Contributions:**

This paper studies the resilience of LSTM architecture, meant as the time required for such model to return into a "normal" state after an "anomalous" input (this is formally defined in Definition 5.1, where two inputs differ until time $t_0$, and it questioned the time $t' > t_0$ such that the outputs of two LSTMs evaluated on these two inputs get back to being $e$ close, where $e$ is some tolerance value).

The Authors derive an upper bound on such time, uniformly on the input space. Technically, they do so via the definition of the invariant sets in Section 4.1. In particular, the important assumption for Theorem 5.4 (upper bound on the recovery time) to hold is that some spectral radii respect $\rho(k) < 1$, for the mapping to be contractive. These spectral radii depend on (and are decreasing in) the value of $k$ which in turn depends on the architecture and the input universe ($x_{\max}$), and the Authors do not make this connection explicit in the Theorem. Rather, the interpretation of the theorem is more like

"If the architecture respects _this condition_ that depends on $k$ (where for $k = 0$ this condition is trivial, and the condition becomes more restrictive as $k$ increases), then the resilience time is upper bounded by _this quantity_, which itself is decreasing in $k$ (Remark 5.5)."

If I understood correctly, the case $k=0$ matches with prior work (Remark 4.3), and the contribution here is to make the upper bound tighter via the definition of the invariant sets for larger $k$. Again, if I understand correctly, it is not obvious how to verify practically what is the largest value of $k$ for which their assumptions hold. In Figure 3, they show that the upper bound obtained for $k = 20$ matches the empirical measure of the resilience time better than the case $k = 0$.

**Audience:**

Yes

**Audience Explanation:**

This is a weak yes in my opinion. The purpose of this work seems rather narrow, and it seems a quite incremental improvement with respect to the works by Terzi et al. 2021 and Bonassi et al 2023. However, my confidence is very low on this regard, as I am not familiar with the literature on control theory and especially on its intersection with architecture-based (here, LSTM) problems. Then, I believe "some individuals" would be interested in the findings of this paper, but not aware how large this sub-community would be.

**Claims And Evidence:**

No

**Claims Explanation:**

My answer is not motivated by the fact that I see incorrect statements in this paper, which is cautiously and carefully written. My main concern is its lack of interpretability and the highly specific theoretical setup. The improvement with respect to past work is encoded in a condition on some invariant sets (indexed by $k$) which is however hard to verify (perhaps as hard as directly measuring the resilience time of the architecture). The authors experimentally see that the bounds become tigher for larger values of $k$, and in the contribution it is stressed that this work considers a data-independent setting. However, I do not see how one can derive data-independent lower bounds on $k$ for the purpose of better resilience time upper bounds with respect to prior work (in the experiments, if I understand correctly, the authors define a specific input sequence so that the Assumptions of Theorem 5.4 are verified for $k = 20$ (see beginning of Section C.1.2).

Beyond that, I find the statements and the theoretical results rather convoluted and not very reader-friendly. The definitions in Section 4.1 and Proposition 5.3 are not easy at all to parse, and there is little to no effort from the side of the Authors to elaborate or give an intuition of the semantic meaning behind the equations.

**Requested Changes:**

Unrelated to the comments above, I would perhaps suggest the authors to include in the introduction the formal definition of recovery time, such as a slimmer version of what appears in Theorem 5.4. As somebody not familiar with this topic, I found the discussion in the introduction too high level to actually grasp the quantity the Authors were interested in.

---

> ### Author Response · Authors · 2026-05-14
> **Response to reviewer SD9P**
>
> Thank you very much for your positive and encouraging review. We appreciate your recognition of the motivation behind our work.
> However, we believe there may be some confusion regarding the role of $k$, and we would therefore like to clarify the following points:
>
> There are two points in your summary that we would like to explain before addressing your remaining comments.
>
> 1. *--(where for $k=0$ this condition is trivial, and the condition becomes more restrictive as $k$ increases)*
>
> For the same parameter values, $\rho(A(k))$ decreases as $k$ increases. Since the sufficient condition for stability is $\rho(A(k)) < 1$, the condition actually becomes less restrictive as $k$ increases.
>
> 2.  *--Again, if I understand correctly, it is not obvious how to verify practically what is the largest value of $k$ for which their assumptions hold.*
>
> Since the spectral radius decreases with respect to $k$, there is no largest value of $k$ for which the condition $\rho(A(k)) < 1$ holds. On the other hand, there may exist a minimum value of $k$ for which the condition is satisfied.
>
> *--My answer is not motivated by the fact that I see incorrect statements in this paper, which is cautiously and carefully written. My main concern is its lack of interpretability and the highly specific theoretical setup. The improvement with respect to past work is encoded in a condition on some invariant sets (indexed by $k$) which is however hard to verify (perhaps as hard as directly measuring the resilience time of the architecture). The authors experimentally see that the bounds become tigher for larger values of $k$, and in the contribution it is stressed that this work considers a data-independent setting. However, I do not see how one can derive data-independent lower bounds on $k$ for the purpose of better resilience time upper bounds with respect to prior work (in the experiments, if I understand correctly, the authors define a specific input sequence so that the Assumptions of Theorem 5.4 are verified for $k=20$ (see beginning of Section C.1.2).*
>
> Let us clarify the role of $k$. This is not a data-dependent tuning parameter, but rather a parameter that controls the accuracy of the invariant set approximation. Verification of the sufficient condition can be carried out using only the network weights and the range of the input set, without requiring a specific input sequence. We consider the input setting to be a practically natural assumption.
> For evaluation, $k$ can be increased as needed. Since $k$ simply represents the number of substitutions into the recurrence relation, the computation can be continued until the value no longer decreases. On the other hand, for training, obtaining a computable desired lower bound on $k$ would be a practically useful indicator; we therefore leave this as future work.
> Nevertheless, because larger $k$ generally leads to better approximations, we believe it is reasonable in practice to choose $k$ based on the available computational cost, as discussed in Remark 4.3(3).
>
> *--Beyond that, I find the statements and the theoretical results rather convoluted and not very reader-friendly. The definitions in Section 4.1 and Proposition 5.3 are not easy at all to parse, and there is little to no effort from the side of the Authors to elaborate or give an intuition of the semantic meaning behind the equations.*
>
> We will add more detailed explanations (e.g., interpretation for the theorem and proof sketch) around Sections 4-5 in the main text.
>
> *--Unrelated to the comments above, I would perhaps suggest the authors to include in the introduction the formal definition of recovery time, such as a slimmer version of what appears in Theorem 5.4. As somebody not familiar with this topic, I found the discussion in the introduction too high level to actually grasp the quantity the Authors were interested in.*
>
> Adding mathematical formulas to the introduction would make it confusing. Therefore, we would like to add a simple definition of recovery time to the caption of Figure 1.

---

### Decision · Action_Editor_W2xF · 2026-06-26

**Recommendation:** Accept with minor revision

**Additional Comments:**

Several technical and presentational problems remain unresolved. First, the claim in Remark 4.3(i) that the $k=0$ case "coincides" with Bonassi et al. [1] is incorrect: the two invariant set constructions differ structurally--the submission explicitly accounts for output gate saturation and separates weight contributions element-wise, making it strictly tighter than Bonassi's concatenated-norm bound even at $k=0$; additionally, the two papers use different LSTM gating conventions, making a direct term-by-term comparison invalid. The authors should replace this claim with an accurate comparison that, incidentally, strengthens (rather than understates) their contribution.

Second, the $\mu^{(l)}(t)$ term is used in the proof of Theorem 4.2 before being formally defined in Proposition 5.3, and the revised manuscript still does not adequately justify treating it as a constant in the multi-layer telescoping argument--the correct justification is that $\sup_t \mu^{(l)}(t)$ is finite because $\mu^{(l)}(t)$ grows at most polynomially while $\rho^t$ decays exponentially, and this should be stated explicitly. Third, the motivational claim that "LSTM surpasses other models in the accuracy of prediction" is supported only by references from 2002, 2015, and 2017 [2, 3, 4]. The paper should instead motivate the work through the continued prevalence of LSTM use in time-series forecasting [9] and safety-critical MPC pipelines [5, 6], while acknowledging that the long-term memory limitations revealed by the additional empirical results constrain the settings in which the theoretical guarantees are practically relevant--a caveat that should be stated explicitly rather than left implicit in the choice of benchmarks. I recommend removing the FPGA sentence entirely; if embedded deployment is to be retained as a motivation it requires updated references, but the forecasting and MPC framing is far more natural given current work. The relationship to Bayer et al. [7] should clarify that the direct justification for $\delta$ISS of the LSTM prediction model in MPC runs through Terzi et al. [8], with Bayer providing the broader NMPC context.

Finally, the long-term memory degradation experiment submitted in the author response must be included in the paper, as it materially affects the interpretation of the method's practical scope. The results show a one-to-two order of magnitude increase in MSE on memorization tasks the moment any resilience penalty is applied ($\lambda > 0$), with no useful operating point between
full memory capability and significant degradation--a substantially sharper trade-off than the two-tank benchmark suggests. Combined with the observation that unpenalized models trained on realistic tasks exhibit spectral radii on the order of $10^2$–$10^3$, far above the $\delta$ISS requirement, these results indicate that the method's practical applicability is currently limited to systems with short effective memory horizons. This is not a fatal limitation, but it must be stated clearly and prominently rather than deferred to a limitations footnote, so that readers can accurately assess the conditions under which the theoretical guarantees are operationally meaningful.

## More (Specific) Requested Edits

- "While typical time-series models can extract nonlinear features from time-series data, store them as internal states, and utilize them to make inferences at each time step, LSTM surpasses other models in the accuracy of prediction (see, e.g., Gers et al., 2002; Ma et al., 2015; Zhao et al., 2017)"--these references are outdated to support this claim. Replace with recent work situating LSTM in the current landscape of safety-critical control and MPC applications, e.g. [5, 6]. For time-series forecasting specifically, LSTMs remain an active area of study; see for example [9].  Note that the primary architecture in [9], P-sLSTM, uses exponential rather than sigmoid gating and is itself outside the scope of the submission's theoretical framework, as detailed in the architecture discussion edit below; [9] should be cited only for the general claim that LSTM-based research remains active, not as a representative application of the submitted theory.

- Remove: "Moreover, LSTM can be implemented on Field Programmable Gate Array (FPGA), which is suitable for machine embedding (Guan et al., 2017)." This is a dated and tangential motivation.  If embedded deployment is to be retained as a motivation, it should be supported with recent literature on LSTM inference on embedded hardware.  A better/easier motivation would be through the prevalence of LSTM use for time series forecasting, although care regarding how well such work requires long-context performance will be necessary (since this was a clear limitation given the additional empirical results).

- Remove the subsection heading and sentence: "We introduce the background, objectives, and contributions of this study. 1.1 Background"

- "risks: If a model" <- lowercase "if": "risks: if a model"

- "We will refine and improve upon these previous studies to achieve the above three goals." <- use active present tense: "Herein, we refine and improve upon these previous studies to achieve the above three goals."

- The bullet "Refinement of an evaluation of LSTM's invariant sets, which form the foundation for stability analysis (Section 4)." will be opaque to general readers who have not yet encountered the concept of an invariant set in this context. Add a parenthetical gloss, e.g. "...invariant sets (bounded regions of state space to which LSTM dynamics are confined under bounded inputs)..."

- Remark 4.3(i) states that the $k=0$ case "coincides with existing results (Terzi et al., 2021; Bonassi et al., 2023)." This is inaccurate with respect to Bonassi et al. [1]: the invariant set constructions differ structurally, and the two papers use different LSTM gating conventions.  Revise to give an accurate comparison, noting that the present paper's bound is strictly tighter even at $k=0$ due to explicit output gate accounting and element-wise weight separation.

- The $\mu^{(l)}(t)$ term appears in the proof of Theorem 4.2 but is formally defined only later in Proposition 5.3. Reorder the exposition so that $\mu^{(l)}(t)$ is defined before it is used, or add a forward reference and explicit statement that $\sup_t \mu^{(l)}(t)$ is finite because $\mu^{(l)}(t)$ grows at most polynomially while $\rho^t$ decays exponentially, justifying its treatment as a bounded constant in the telescoping argument.

- Section 2.1 cites Bayer et al. [7] as direct justification for why $\delta$ISS of the LSTM model matters in MPC. Clarify that Bayer's result concerns $\delta$ISS of the plant in a tube-MPC setting, and that the direct justification for $\delta$ISS of the LSTM prediction model specifically runs through Terzi et al. [8].

- The long-term memory degradation results and spectral radius data provided in the author response to Reviewer 3 must be incorporated into the paper--either as a dedicated subsection within the experiments or as a clearly labelled addition to the limitations section--with the accompanying tables. These results materially bound the practical scope of the method and should not remain only in review correspondence.

- The limitations section should be revised to state explicitly and prominently that the method's applicability is currently restricted to systems with short effective memory horizons, as evidenced by the one-to-two order of magnitude MSE degradation observed under any resilience penalty in the memorization experiments, and by the fact that unpenalized models on realistic tasks exhibit spectral radii of order $10^2$-$10^3$, far outside the $\delta$ISS regime.

- The submission must include a discussion of the relationship between its theoretical framework and current high-impact LSTM architectures. The sigmoid-gated LSTM analyzed in the submission is no longer the dominant architecture in time-series forecasting or language modeling. The most relevant recent developments are: (i) sLSTM and mLSTM within the xLSTM family (Beck et al., 2024 [10]), which introduce exponential gating and, in the case of mLSTM, a fully parallelizable matrix memory with a covariance update rule; (ii) P-sLSTM (Kong et al., 2025 [9]), which combines sLSTM with patching and channel independence and achieves state-of-the-art forecasting results; and (iii) the continuing use of sigmoid-gated LSTM and GRU in safety-critical embedded control and MPC [5, 6, 11, 12], where the standard architecture remains in active deployment precisely because its stability properties are better understood. The submission's results apply to category (iii) but not (i) or (ii). The theoretical framework covers sigmoid-gated LSTM as defined in equations (2a)--(2d); the extension to exponential-gated variants is precluded by three specific failures: the Lipschitz constant of $1/4$ used throughout Theorems 4.2 and A.5 does not exist globally for the exponential function; the invariant set bound $c^{(l)}(k)$ has no finite value when the forget gate output can exceed 1; and the two-variable state-space formulation $(c^{(l)}, h^{(l)})$ does not accommodate the normalizer state $n_t$ of sLSTM, which introduces a division $c_t/n_t$ that is not Lipschitz near zero. Notably, Kong et al. [9] prove via a geometric ergodicity argument (following Zhao et al., 2020) that sLSTM also has short memory under the same contractivity condition that would be required for $\delta$ISS--their Case 1 and the submission's Theorem 4.2 are structurally analogous results in adjacent theoretical traditions (Markov chain ergodic theory vs. control-theoretic ISS), converging on the same qualitative conclusion. The submission's quantitative recovery time bound has no counterpart in Kong et al., which is the clearest statement of what is genuinely novel here.  The extension of the $\delta$ISS framework and recovery time analysis to exponential-gated architectures should be identified as a concrete open problem.

## References

[1] F. Bonassi, A. La Bella, G. Panzani, M. Farina, and R. Scattolini, "Deep Long-Short Term
Memory networks: Stability properties and Experimental validation," in *2023 European Control
Conference (ECC)*, Bucharest, Romania, 2023, pp. 1–6.

[2] F. A. Gers, N. N. Schraudolph, and J. Schmidhuber, "Learning Precise Timing with LSTM
Recurrent Networks," *Journal of Machine Learning Research*, vol. 3, pp. 115–143, 2002.

[3] X. Ma, Z. Tao, Y. Wang, H. Yu, and Y. Wang, "Long short-term memory neural network for
traffic speed prediction using remote microwave sensor data," *Transportation Research Part C:
Emerging Technologies*, vol. 54, pp. 187–197, 2015.

[4] Z. Zhao, W. Chen, X. Wu, P. C. Y. Chen, and J. Liu, "LSTM network: a deep learning approach for short-term traffic forecast," *IET Intelligent Transport Systems*, vol. 11, pp. 68–75, 2017.

[5] Z. Wu, A. Tran, D. Rincon, and P. D. Christofides, "Machine learning-based predictive control of nonlinear processes. Part I: Theory," *AIChE Journal*, vol. 65, p. e16729, 2019.

[6] M. Jung, P. R. da Costa Mendes, M. Önnheim, and E. Gustavsson, "Model Predictive Control when utilizing LSTM as dynamic models," *Engineering Applications of Artificial Intelligence*, vol. 123, p. 106226, 2023.

[7] F. Bayer, M. Bürger, and F. Allgöwer, "Discrete-time Incremental ISS: A framework for Robust NMPC," in *2013 European Control Conference (ECC)*, Zurich, Switzerland, 2013, pp. 2068–2073.

[8] E. Terzi, F. Bonassi, M. Farina, and R. Scattolini, "Learning model predictive control with long short-term memory networks," *International Journal of Robust and Nonlinear Control*, vol. 31, no. 18, pp. 8877–8896, 2021.

[9] Kong, Yaxuan, et al. "Unlocking the power of lstm for long term time series forecasting." Proceedings of the AAAI conference on artificial intelligence. Vol. 39. No. 11. 2025.

[10] M. Beck, K. Pöppel, M. Spanring, A. Auer, O. Prudnikova, M. Kopp, G. Klambauer, J. Brandstetter, and S. Hochreiter, "xLSTM: Extended Long Short-Term Memory," *arXiv preprint arXiv:2405.04517*, 2024.

[11] F. Bonassi, M. Farina, J. Xie, and R. Scattolini, "On recurrent neural networks for learning-based control: Recent results and ideas for future developments," *Journal of Process Control*, vol. 114, pp. 92--104, 2022.

[12] W. D'Amico, A. La Bella, M. Farina, and L. Zaccarian, "Regional stability conditions for recurrent neural network-based control systems," *Automatica*, vol. 174, p. 112127, 2025.

**Audience:**

Yes

**Audience Explanation:**

LSTMs remain widely used for time-series forecasting. Upon completion of the requested edits, the theoretical results will be solid for $\sigma$-gated LSTMs, opening the door to extending related results to alternative LSTM architectures in active use.

**Claims And Evidence:**

Yes

**Claims Explanation:**

The majority of reviewers are in favor of accept, and the submission makes a genuine theoretical contribution by introducing recovery time as a formal metric for LSTM resilience and deriving a computable, data-independent upper bound via incremental input-to-state stability theory. The theoretical framework is sound, the invariant set analysis is carefully constructed, and the layer-wise tightening through the parameter $k$ is a meaningful improvement over prior work. The resilience-aware training procedure translates the theory into a practical loss penalty, and the author responses during review clarified several misunderstandings about the role of $k$ and resolved legitimate questions about computational overhead. The outstanding concern from Reviewer 2 regarding the forward reference to $\mu^{(l)}(t)$ and its treatment as a constant in the telescoping argument is not fully resolved in the revised manuscript and remains an open presentation issue.  However, the requested edits (see below) include the correct justification.

I am recommending acceptance pending several important requested revisions.

---

> ### Author Response · Authors · 2026-07-08
> **Clarification Regarding Theorem 4.2 and Minor Errors in the Revised Manuscript**
>
> Thank you very much for accepting our manuscript, and for your valuable comments. We are currently revising it in accordance with your comments, and would appreciate your patience until the deadline of July 25. We would like to raise one question and report two minor issues, as detailed below.
>
> Regarding the point that Theorem 4.2 for the case of k=0 does not coincide with the results of Bonassi et al. (2023) or Terzi et al. (2019), we have identified differences in the choice of invariant sets and norms. However, we could not identify any substantive difference in the LSTM gate structures themselves — it seems to us that the differences lie only in the notation used across the papers. To make the comparison easier, we have summarized the correspondence of notation in the table below. If our understanding is correct, the gate structures should be equivalent once the notation is aligned, but please let us know if we have overlooked something.
>
> ### Notation of LSTM. Note that our model and the model in Bonassi et al. (2023) are multi-layer, whereas the model in Terzi et al. (2021) is single-layer. In addition, both $\sigma$ and $\sigma_g$ denote the sigmoid function, and both $\phi$ and $\sigma_c$ denote the hyperbolic tangent function.
>
> | Name | Ours | Bonassi et al., 2023 | Terzi et al., 2021 |
> |---|---|---|---|
> | Time Index | t | k | k |
> | Cell State | $c^{(l)}(t)$ | $c^{(l)}_k$ | $x$ |
> | Hidden State | $h^{(l)}(t)$ | $h^{(l)}_k$ | $\xi$ |
> | Input | $x^{(l)}(t)$ | $u^{(l)}_k$ | $u$ |
> | Forget Gate | $\sigma(W^{(l)}_f x^{(l)}(t) + U^{(l)}_f h^{(l)}(t) + b^{(l)}_f)$ | $f^{(l)}_k=\sigma(W^{(l)}_f u^{(l)}_k + U^{(l)}_f h^{(l)}_k + b^{(l)}_f)$ | $\sigma_g (W_f u + U_f \xi + b_f)$ |
> | Input Gate | $\sigma(W^{(l)}_i x^{(l)}(t) + U^{(l)}_i h^{(l)}(t) + b^{(l)}_i)$ | $i^{(l)}_k=\sigma(W^{(l)}_i u^{(l)}_i + U^{(l)}_i h^{(l)}_i + b^{(l)}_i)$ | $\sigma_g (W_i u + U_i \xi + b_i)$ |
> | Output Gate | $\sigma(W^{(l)}_o x^{(l)}(t) + U^{(l)}_o h^{(l)}(t) + b^{(l)}_o)$ | $z^{(l)}_k=\sigma(W^{(l)}_z u^{(l)}_k + U^{(l)}_z h^{(l)}_k + b^{(l)}_z)$ | $\sigma_g (W_o u + U_o \xi + b_o)$ |
> | Squashed Input | $\phi(W^{(l)}_c x^{(l)}(t) + U^{(l)}_c h^{(l)}(t) + b^{(l)}_c)$ | $r^{(l)}_k=\phi(W^{(l)}_r u^{(l)}_k + U^{(l)}_r h^{(l)}_k + b^{(l)}_r)$ | $\sigma_c (W_c u + U_c \xi + b_c)$ |
>
> In addition, we would like to report two errors we noticed after submitting the revised manuscript. These are unrelated to your comments.
>
> First, the caption of the table presenting the experimental results on long-term knowledge was incorrect:
>
> Average MSE (at step 2000) over 3 samples → Average MAE (at step 2000) over 3 samples
>
> The values in the table itself were computed as MAE from the beginning; only the caption incorrectly stated MSE.
>
> Second, we inserted "Section D: Adversarial Training" between Sections 7.2.1 and 7.2.2. As a result, Section 7.2.2 was renumbered to 8.2.2, which now appears in an unnatural position within the manuscript.
>
> We correct them when submitting the camera-ready version. Please let us know if you have any questions or concerns regarding this, and we will be happy to address them.
>
> Thank you for your time and consideration.

---

> > ### Author Response · Authors · 2026-07-15
> > **Correction to my previous message (MSE/MAE table caption)**
> >
> > Please disregard my previous message about the table caption — I had it backwards.
> > ~~Average MSE (at step 2000) over 3 samples → Average MAE (at step 2000) over 3 samples~~
> > ~~The values in the table itself were computed as MAE from the beginning; only the caption incorrectly stated MSE.~~
> > After re-checking the evaluation code, the values in the table are in fact MSE, so the original caption "Average MSE (at step 2000) over 3 samples" was correct as written. The numerical values are unchanged. I apologize for the confusion.

---

> ### Comment · Action_Editor_W2xF · 2026-07-20
> **Follow up**
>
> Thank you for the careful notation comparison. The authors are correct that the gate structures
> are equivalent up to notation, and my earlier comment was imprecise in claiming a difference in
> gating conventions. That language should be read as referring to the variable naming rather than
> any substantive architectural difference (thanks to the authors for clarifying).
>
> However, the following point remains: the difference lies in the invariant set construction and
> norms, as the authors identified. Specifically, the submission's bound is strictly tighter than
> Bonassi et al. (2023) even at $k = 0$ for two reasons:
>
> 1. The submission separates the weight, recurrent, and bias contributions element-wise using
> the positive-part operator, rather than using Bonassi's concatenated joint norm over all three
> matrices together, which is a coarser bound.
>
> 2. The submission's hidden state bound explicitly incorporates the output gate saturation
> factor $\sigma_o(k)$, giving a bound of the form $\eta(k) = \varphi(c(k)) \cdot \sigma_o(k)$,
> whereas Bonassi bounds the hidden state by $\varphi(\bar{c})$ alone. Since $\sigma_o(k) < 1$,
> the submission's bound is strictly smaller.
>
> Remark 4.3(i) should therefore be revised to replace the coincidence claim with a statement
> that the submission's $k = 0$ bound is already strictly tighter than Bonassi et al. (2023) due
> to these two structural differences in the invariant set construction, and that the $k > 0$
> sequence tightens it further. This is a stronger contribution statement than the current
> remark, and the authors are encouraged to frame it as such in the final draft.
>
> No worries about the caption.